# Understanding Multimodal LLMs Under Distribution Shifts: An Information-Theoretic Approach

Changdae Oh [1]   Zhen Fang [2]   Shawn Im [1]   Xuefeng Du [1]   Yixuan Li [1]

## Abstract

Multimodal large language models (MLLMs) have shown promising capabilities but struggle under distribution shifts, where evaluation data differ from instruction tuning distributions. Although previous works have provided empirical evaluations, we argue that establishing a formal framework that can characterize and quantify the risk of MLLMs is necessary to ensure the safe and reliable application of MLLMs in the real world. By taking an information-theoretic perspective, we propose the first theoretical framework that enables the characterization of the maximum risk of MLLMs under distribution shifts. Central to our framework is the introduction of Effective Mutual Information (EMI), a principled metric that quantifies the relevance between input queries and model responses. We then derive an upper bound for the EMI difference between in-distribution (ID) and out-of-distribution (OOD) data, connecting it to visual and textual distributional discrepancies. Extensive experiments on real benchmark datasets, spanning 61 shift scenarios, empirically validate our theoretical insights.     Code: 

## 1. Introduction

Multimodal large language models (MLLMs) have demonstrated remarkable capabilities in handling complex tasks that require reasoning over both visual and textual modalities. By leveraging visual instruction tuning (Liu et al., 2023; Dai et al., 2023; Zhu et al., 2024), MLLMs have shown promise in answering open-ended questions and generating contextually relevant captions. As a critical aspect for real-world deployment, MLLMs are expected to operate robustly in the wild, where *distribution shifts* occur—that is, when the evaluation data deviates from the instruction tuning data, whether due to changes in visual inputs (e.g, domain-specific images), textual inputs (e.g., linguistic variations), or the combination thereof. However, negative reports on MLLM failures in edge cases have steadily emerged, raising concerns about their reliability.

For example, MLLMs struggle with queries in specialized domains such as medicine and chemistry (Zhang et al., 2024a; Han et al., 2024; Zhou et al., 2024), perform poorly on simple image classification tasks compared to open-ended question answering (Zhang et al., 2024b; Zhai et al., 2024), and exhibit hallucinations to biased queries (Li et al., 2023b; Ye-Bin et al., 2025). Given the increasing impact of MLLMs, it is crucial to understand their failure modes under distribution shifts. Despite the significance of the problem, existing works often lack a fine-grained diagnosis for various factors of shifts. More importantly, the absence of a formal framework to explain the underlying principle further hinders the systematic understanding of an MLLM's behavior. This motivates us to raise the research question:

***Could we derive a theoretical framework to characterize MLLM's behavior under distribution shifts?***

To address this, we propose an information-theoretic framework that characterizes MLLM performance under distribution shifts with theoretical rigor and practical interpretability. Our framework is well-suited for analyzing instruction-tuned models, wherein the purpose of learning is naturally connected to the mutual information between the input query and response. In this framework, we introduce *effective mutual information* (EMI) as a principled measure to access the relevance between an input query and model response. Intuitively, EMI is expected to be higher when a test input query originates from the in-distribution (ID) data, similar to those used during instruction tuning, compared to when the input comes from out-of-distribution (OOD). To quantify this performance gap, we compute the difference in EMI between ID and OOD and derive an upper bound for this difference, expressed in terms of distributional discrepancies in input and output spaces (see Theorem 4.5 and 4.6). By grounding the measure in information theory, we provide

[1]Department of Computer Sciences, University of Wisconsin–Madison, WI, USA [2]Faculty of Engineering & Information Technology, University of Technology Sydney, Sydney, Australia. Correspondence to: Yixuan Li <sharonli@cs.wisc.edu>.

*Proceedings of the 42$^{nd}$ International Conference on Machine Learning*, Vancouver, Canada. PMLR 267, 2025. Copyright 2025 by the author(s).

the first theoretical framework to analyze and understand the impact of distribution shifts on MLLM performance.

Beyond the theoretical rigor, we further show that our framework holds practical value. In particular, we demonstrate that EMI is closely related to a widely used empirical metric for MLLMs, relative preference score under the LLM-as-a-judge paradigm (Zheng et al., 2023). The relative preference score commonly relies on an external judge model, e.g., GPT-4o (Hurst et al., 2024), thus it lacks mathematical guarantees due to the black-box nature of these models. In contrast, EMI provides an *theory-grounded measurement* of the relevance between input queries and output responses of the MLLM being evaluated, offering a fundamental basis for assessing the MLLM's performance gap under shifts.

Finally, we conduct comprehensive validation for the theoretical framework and show that our theorems empirically hold in real-world benchmarks. Our experiments comprehensively examine 34 synthetic and 27 natural distribution shift scenarios, resulting in a total number of 61 ID-OOD evaluations for each MLLM. Results confirm strong correlations between empirical estimates of EMI and relative preference, as well as correlations between the EMI difference and its upper bounds, demonstrating the effectiveness of our framework in capturing performance gaps under diverse shifts. Our contributions can be summarized as follows:

- We propose a new framework, effective mutual information (EMI), to analyze MLLM under distribution shift, and justify the use of EMI by showing the theoretical connection between EMI and the LLM-judge-driven relative preference score.

- We derive theoretical upper bounds of MLLM's performance gap, which can be characterized by shifts over multimodal input queries and output discrepancies.

- We empirically verify our theoretical statements on 61 real-world distribution shift scenarios of open-ended question-answering benchmarks with six MLLMs.

## 2. Preliminary

**Random variable and distribution.** Let $\mathcal{X} = \mathcal{X}_v \times \mathcal{X}_t$ denote the input space, where $\mathcal{X}_v$ and $\mathcal{X}_t$ correspond to the visual and textual feature spaces, respectively. Similarly, let $\mathcal{Y}$ denote the response space. We define the random variables $\mathbf{X} = (X_v, X_t) \in \mathcal{X}$ and $Y \in \mathcal{Y}$, where $\mathbf{X}$ is the sequence of tokens that combine visual and text input queries, and $Y$ represents the associated response tokens. The joint distribution is denoted by $P_{\mathbf{X}Y}$, with marginals $P_{\mathbf{X}}$, $P_Y$, and the conditional distribution $P_{Y|\mathbf{X}}$. For subsequent sections, $P_{\mathbf{X}Y}$ refers to the instruction tuning data distribution, which we consider as in-distribution (ID).

**MLLM and visual instruction tuning.** MLLM usually consists of three components: (1) a visual encoder, (2) a vision-to-language projector, and (3) an LLM backbone that processes a multimodal input sequence to generate a valid textual output $y$ in response to an input query $\mathbf{x}$. An MLLM can be regarded as modeling a conditional distribution $P_\theta(y|\mathbf{x})$, where $\theta$ is the model parameters. To attain the multimodal conversation capability, MLLMs commonly undergo a phase, so-called *visual instruction tuning* (Liu et al., 2023), with a conditional language modeling loss:

$$\arg\min_{\theta \in \Theta} \mathbb{E}_{\mathbf{x},y \sim P_{\mathbf{X}Y}} [\sum_{l=1}^{L} -\log P_\theta(y_l|\mathbf{x}, y_{<l})], \quad (1)$$

where $L$ is a sequence length and $y = (y_1, ..., y_L)$. After being trained by Eq. (1), MLLM produces a response given a query of any possible tasks represented by text.

**Evaluation of open-ended generations.** (M)LLM-as-a-judge method (Ouyang et al., 2022; Zheng et al., 2023; Kim et al., 2024) is commonly adopted to evaluate open-ended generation. In this paradigm, a judge model produces preference scores or rankings for the responses given a query, model responses, and a scoring rubric. Among the evaluation metrics, the *relative preference score* (RP score; Eq. (2)) is one of the most representative ones.

**Definition 2.1** (**Relative Preference Score**). *Given a reward function* $r : \mathcal{X} \times \mathcal{Y} \to \mathbb{R}$, *the relative preference (RP) score of model* $P_\theta$ *w.r.t.* $P_{\mathbf{X}Y}$ *are defined as follows:*

$$RP(P_{\mathbf{X}Y}; P_\theta) := \mathbb{E}_{\substack{\mathbf{x},y \sim P_{\mathbf{X}Y} \\ \hat{y} \sim P_\theta(\cdot|\mathbf{x})}} [r(\mathbf{x}, \hat{y})/r(\mathbf{x}, y)]. \quad (2)$$

Here, the reward function $r$ can be any possible MLLMs, such as GPT-4o (Hurst et al., 2024) or discriminative language models (Lambert et al., 2024), and we often take an output from another MLLM (usually more powerful than $P_\theta$) as a reference answer $y$.

## 3. Motivation

**A systematic understanding of MLLM under distributional shifts.** While instruction-following MLLMs are designed to handle a diverse range of tasks, they often struggle with specialized domains (Zhang et al., 2024a; Zhou et al., 2024), perform poorly on simple image classification tasks (Zhai et al., 2024; Zhang et al., 2024b), and hallucinate to biased queries (Li et al., 2023b; Ye-Bin et al., 2025). We argue that the fundamental cause of these failure modes in MLLMs can be traced back to distribution shifts. Specifically, poor performance on classification tasks and specialized distributions can be attributed to shifts between instruction tuning distribution $P_{\mathbf{X}Y}$ and evaluation distribution $Q_{\mathbf{X}Y}$. This work comprehensively analyzes three types of distributional shifts that can arise in MLLM as follow:

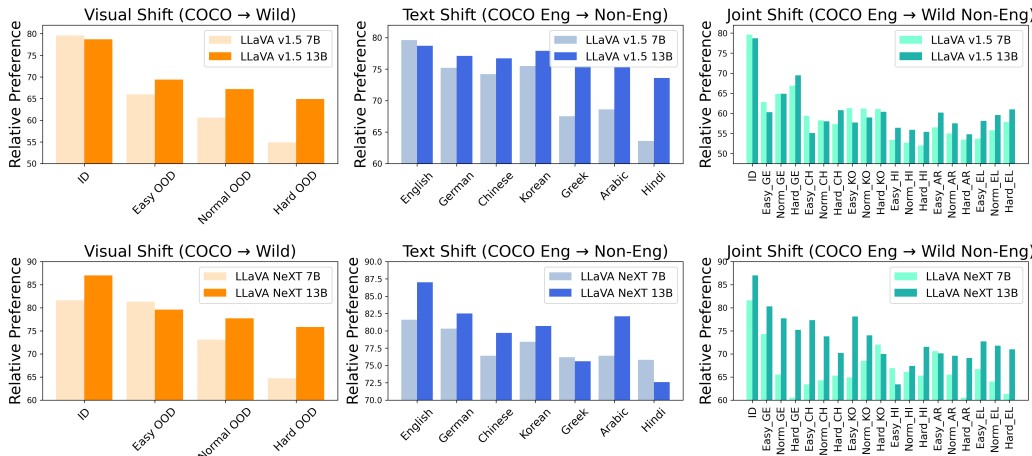

*Figure 1.* **Performance variation against varying distribution shifts.** We evaluated LLaVA v1.5 (top) and LLaVA NeXT (bottom) models on 27 out-of-distribution (OOD) variants of the LLaVA-Bench COCO (ID). Here, the $x$-axis is sorted by the severity of shifts between ID and OOD. There is a consistent trend, increased degrees of distribution shifts result in performance degradations of MLLM.

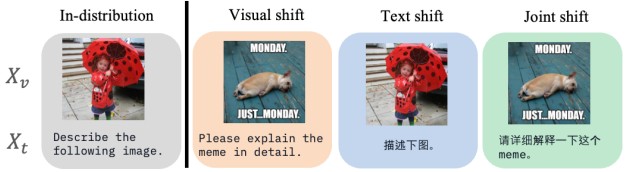

*Figure 2.* **Types of distribution shifts between train and evaluation of MLLMs.** We simulate visual, text, and joint shifts by controlling the shift of each input modality.

1. *Visual shift*: the marginal distribution of visual query undergoes shift $D(P_{X_v} \| Q_{X_v}) \gg 0$, while that of text query remains largely unchanged $D(P_{X_t} \| Q_{X_t}) \approx 0$.

2. *Text shift*: the marginal distribution of text query undergoes shift $D(P_{X_t} \| Q_{X_t}) \gg 0$, while that of visual query remains largely unchanged $D(P_{X_v} \| Q_{X_v}) \approx 0$.

3. *Joint shift*: both visual and text queries suffer shifts simultaneously, and the relationship between visual and text queries may also shift $D(P_{\mathbf{X}} \| Q_{\mathbf{X}}) \gg 0$,

where $D$ denotes a divergence that measures the discrepancy between distributions $P$ and $Q$. For $M = (P + Q)/2$, one can measure the Kullback-Leibler (KL) divergence and Jensen-Shannon (JS) divergence as below:

$$D_{\mathrm{KL}}(P \| Q) = \mathbb{E}_{\mathbf{z} \sim P}[\log P(\mathbf{z})/Q(\mathbf{z})],$$
$$D_{\mathrm{JS}}(P \| Q) = [D_{\mathrm{KL}}(P \| M) + D_{\mathrm{KL}}(Q \| M)]/2.$$

**Pilot study.** We hypothesize that: (1) performance degradation in MLLMs becomes more severe as $Q_{\mathbf{X}Y}$ deviates further from the $P_{\mathbf{X}Y}$; (2) the amount of total performance degradation can be factored into visual query shift and text query shift. To test these hypotheses, we design three types of shifts—visual shift, text shift, and joint shift—illustrated in Figure 2, and evaluate MLLMs under these shifts.

Specifically, we adopt LLaVA-1.5 (Liu et al., 2023) and LLaVA-NeXT (Liu et al., 2024a) in 7B and 13B sizes as our target MLLM, with LLaVA-Bench COCO (Liu et al., 2023) serving as the ID dataset, which is distributionally similar to the instruction tuning data. We adopt LLaVA-Bench Wild (Liu et al., 2023) to vary visual input semantics, and we apply language translation with GPT-4, e.g., from English to {Chinese, German, Chinese, Korean, Hindi, Arabic, and Greek}, to realize shifts in text query. We vary the severity of shifts by controlling the magnitude of perturbations in synthetic shift setup and partitioning a dataset based on the mean embedding distance from ID samples in natural shift setup. Following Liu et al. (2023), we evaluate the performance using RP score (Eq. (2)) with GPT-4 judge.

Figure 1 shows the performance variations of MLLMs under different types and magnitudes of distribution shifts, where the $x$-axis is sorted by the severity of shifts (more results from different types of shifts can be founded in Appendix C). Across all models, a consistent trend emerges: as the severity of the shift increases, the performance degradation becomes more significant. This trend robustly holds for both visual and text shifts. Joint shifts result in greater performance degradation, suggesting a complementary effect of shifts across modalities. *These consistent observations suggest that there might exist an underlying principle explaining the relationship between performance variation and distributional discrepancy, which motivates us to investigate the theoretical model behind these empirical results.*

**Our position.** Although there have been similar observations on the performance degradation of MLLM under distribution shifts (Achiam et al., 2023; Zhang et al., 2024a; Zhou et al., 2024; Zhang et al., 2024b), all of them present only the coarse empirical evaluation results without finer analysis on the underlying factor of those performance degradations. To the best of our knowledge, there is no formal framework

to explain the performance variations of MLLMs in terms of distribution shifts— despite its crucial importance for ensuring reliable applications of MLLMs. To bridge this gap, we propose the *first theoretical framework that characterizes MLLM performance variations under distribution shifts from an information-theoretic perspective*.

# 4. Information-theoretic Analysis on MLLM Performance Gap

In this section, we start with introducing the mutual information (MI) and its limitation as a metric in Sec. 4.1, and present a new metric on MLLM evaluation (Sec. 4.2). Then, we derive theorems based on it to characterize the MLLM performance gap under distribution shifts (Sec. 4.3).

## 4.1. Mutual Information for MLLM

A fundamental capability of MLLMs is their instruction-following property (Ouyang et al., 2022)—a direct outcome of instruction-tuning, where the model is trained to generate responses that are aligned with the intent of a given input query or instruction. To evaluate instruction-following capability, we first consider the *mutual information* (Shannon, 1948) to measure the shared information between the query and the corresponding model response.

**Definition 4.1** (**Mutual Information (MI)**). *For a joint distribution $P_{\mathbf{X}Y}$ over $\mathcal{X} \times \mathcal{Y}$, the mutual information with respect to $P_{\mathbf{X}Y}$ is defined as,*

$$I(P_{\mathbf{X}Y}) := \mathbb{E}_{\mathbf{x},y \sim P_{\mathbf{X}Y}}[\log \frac{P_{\mathbf{X}Y}(\mathbf{x}, y)}{P_{\mathbf{X}}(\mathbf{x})P_Y(y)}]. \quad (3)$$

MI is deeply related to the entropy, which is defined as $H(P_{\mathbf{X}}) := -\mathbb{E}_{\mathbf{x} \sim P_{\mathbf{X}}}[\log P_{\mathbf{X}}(\mathbf{x})]$. It is easy to check that $I(P_{\mathbf{X}Y}) = H(P_Y) - \mathbb{E}_{\mathbf{x} \sim P_{\mathbf{X}}}[H(P_{Y|\mathbf{X}=\mathbf{x}})]$. Intuitively, MI captures how much the response tells us about the query and vice versa. One reason for considering MI for model evaluation is that the conditional language modeling objective in Eq. (1) is closely connected to MI as shown in Eq. (4).

**Information-theoretic interpretation of instruction tuning.** In Eq. (4), we show that the objective for instruction tuning (Eq. 1) forms a lower bound of the MI between $\mathbf{X}$ and $Y$ subtracted by entropy over $Y$, given a sufficiently large representation capacity of the model, i.e., small $\delta$. As we are interested in measuring the input-output dependency, we focus on the $I(P_{\mathbf{X}Y})$ term as a metric to effectively gauge the upper bound of visual instruction tuning objective, Eq. (1).

$$\begin{aligned} I(P_{\mathbf{X}Y}) - H(P_Y) &= \mathbb{E}_{\mathbf{x},y \sim P_{\mathbf{X}Y}}[\log P_\theta(y|\mathbf{x})] + \delta \\ &\geq \mathbb{E}_{\mathbf{x},y \sim P_{\mathbf{X}Y}}[\log P_\theta(y|\mathbf{x})], \end{aligned} \quad (4)$$

where $\delta = \mathbb{E}_{\mathbf{x} \sim P_{\mathbf{X}}}[D_{\mathrm{KL}}(P_{Y|\mathbf{X}=\mathbf{x}} \| P_\theta(\cdot|\mathbf{x}))]$.

**Going from instruction tuning to test-time MI.** While $I(P_{\mathbf{X}Y})$ measures the MI between the input query and ground truth response from $P_{\mathbf{X}Y}$, one may be interested in measuring MI between the query and model response on inference-time distributions. We use a tensor product $P_{\mathbf{X}} \otimes P_\theta$ to present a joint distribution between the input distribution $P_{\mathbf{X}}$ and *model output* distribution $P_\theta(y|\mathbf{x})$:

$$P_{\mathbf{X}} \otimes P_\theta := P_{\mathbf{X}}(\mathbf{x})P_\theta(y|\mathbf{x}), \ \forall (\mathbf{x}, y) \in \mathcal{X} \times \mathcal{Y}. \quad (5)$$

Accordingly, the mutual information w.r.t. the joint distribution $P_{\mathbf{X}} \otimes P_\theta$ can be written as:

$$\begin{aligned} I(P_{\mathbf{X}} \otimes P_\theta) = {} & H(\mathbb{E}_{\mathbf{x} \sim P_{\mathbf{X}}}[P_\theta(\cdot|\mathbf{x})]) \\ & - \mathbb{E}_{\mathbf{x} \sim P_{\mathbf{X}}}[H(P_\theta(\cdot|\mathbf{x}))]. \end{aligned} \quad (6)$$

**Limitation of test-time MI under distribution shifts.** Although one could directly use $I(P_{\mathbf{X}} \otimes P_\theta)$, the mutual information between the input query and the model response, as a metric, the vanilla MI may not be suitable for scenarios involving distribution shifts. For example, consider the distribution $P_{\mathbf{X}}$ (e.g., general domain), and the distribution $Q_{\mathbf{X}}$ (e.g., medical domain). Suppose the MI w.r.t. model $P_\theta$ on $P_{\mathbf{X}}$, i.e., $I(P_{\mathbf{X}} \otimes P_\theta)$ is 2.0, while on the $Q_{\mathbf{X}}$, it is $I(Q_{\mathbf{X}} \otimes P_\theta) = 1.0$. Does this imply that model $P_\theta$ performs twice as poorly on $Q_{\mathbf{X}}$? The answer is unclear.

The challenge lies in the inherent variability of MI scales across data domains. Recall the formulation of $I(P_{\mathbf{X}} \otimes P_\theta)$ in Eq. (6), the first term $H(\mathbb{E}_{\mathbf{x} \sim P_{\mathbf{X}}}[P_\theta(\cdot|\mathbf{x})])$ represents the upper bound of MI and varies with the data domain; and the second term $\mathbb{E}_{\mathbf{x} \sim P_{\mathbf{X}}}[H(P_\theta(\cdot|\mathbf{x}))]$ reflects the input-output dependency and depends on the true data-generating process. For instance, given a fixed vocabulary, the responses from MLLM could contain more diverse words in a general domain, whereas a narrower subset of words could be expected in the specialized domains, such as medical. Then, $H(\mathbb{E}_{\mathbf{x} \sim P_{\mathbf{X}}}[P_\theta(\cdot|\mathbf{x})])$ and $\mathbb{E}_{\mathbf{x} \sim P_{\mathbf{X}}}[H(P_\theta(\cdot|\mathbf{x}))]$ would be larger in a general domain. Therefore, we argue that *a desired evaluation metric should disentangle the pure input-response relevance from the intrinsic characteristics of the dataset*.

## 4.2. Effective Mutual Information for Reliable MLLM Evaluation

To remove the influence of the domain-dependent scale, we propose *effective mutual information* (EMI) as a remedy.

**Definition 4.2** (**Effective Mutual Information (EMI)**). *Given a joint distribution $P_{\mathbf{X}Y}$ and an MLLM $P_\theta$ parameterized with $\theta$, the effective mutual information between the input and model response is defined as below,*

$$EMI(P_{\mathbf{X}Y}; P_\theta) := I(P_{\mathbf{X}} \otimes P_\theta) - I(P_{\mathbf{X}Y}). \quad (7)$$

Compared to the standard MI, $EMI(P_{\mathbf{X}Y}; P_\theta)$ measures the "effective" relevance between the query $\mathbf{x}$ and the model

response $\hat{y}$ by subtracting a ground truth MI $I(P_{\mathbf{X}Y})$ from $I(P_{\mathbf{X}} \otimes P_\theta)$. Refer to Figure 5 in Appendix A, for an intuitive example: by accounting for a baseline level of MI, EMI quantifies the extent to which the model captures the effective relevance between the input and output. The use of EMI as an evaluation metric for MLLMs can be further supported by (1) its analogy to the excess risk and effective robustness; (2) its connection to an LLM-judge score.

**Analogy to the excess risk and effective robustness.** The minimum achievable error varies depending on the data-generating process. To enable reliable model selection that is agnostic to data distributions, excess risk—defined as the difference between a model's risk and the minimum possible risk—has been extensively studied (Castro & Nowak, 2008; Koltchinskii, 2010; Mohri, 2018). More recently, Taori et al. (2020) introduced the concept of effective robustness to quantify the "effective" OOD generalization accuracy of classification models by subtracting their ID accuracy. The motivation behind EMI aligns with these concepts, i.e., mitigating the influence of external confounding effects that hinder the accurate measure of model performance. EMI ensures that the metric focuses on the model's effective ability to capture input-output relevance, independent of confounding effects from the data domain.

**Connection to LLM-judge score.** We also show that EMI is closely related to an LLM-judge-based metric, i.e., RP score in Eq. (2), a common metric used to assess MLLM outputs. Conceptually, EMI quantifies the effective relevance between the input query and the model's response by accounting for the baseline mutual information, whereas the RP score measures the relative preference of the model's responses over a reference response. More formally, their connection can be mathematically established through the lens of a logit Bradley-Terry preference model (PM) formulation (Bradley & Terry, 1952; Hunter, 2004), $\mathrm{logit} P(\hat{y} \succ y | \mathbf{x})$, an alternative formulation of relative preference. To be specific, we commonly use $\log P(\hat{y} \succ y | \mathbf{x})$ to train a reward model (RM) which is adopted to compute LLM-judge score, such as Eq. (2). We compare both terms in the following.

$$
\begin{aligned}
\mathrm{PM}(P_{\mathbf{X}Y}; P_\theta) &:= \mathbb{E}_{\substack{\mathbf{x},y \sim P_{\mathbf{X}Y} \\ \hat{y} \sim P_\theta(\cdot|\mathbf{x})}} [\mathrm{logit}\, P(\hat{y} \succ y | \mathbf{x})] \\
&= \mathbb{E}_{\substack{\mathbf{x},y \sim P_{\mathbf{X}Y} \\ \hat{y} \sim P_\theta(\cdot|\mathbf{x})}} [r(\mathbf{x}, \hat{y}) - r(\mathbf{x}, y)], \\
\mathrm{RM}(P_{\mathbf{X}Y}; P_\theta) &:= \mathbb{E}_{\substack{\mathbf{x},y \sim P_{\mathbf{X}Y} \\ \hat{y} \sim P_\theta(\cdot|\mathbf{x})}} [\log\, P(\hat{y} \succ y | \mathbf{x})] \\
&= \mathbb{E}_{\substack{\mathbf{x},y \sim P_{\mathbf{X}Y} \\ \hat{y} \sim P_\theta(\cdot|\mathbf{x})}} [\log \sigma(r(\mathbf{x}, \hat{y}) - r(\mathbf{x}, y))],
\end{aligned}
$$

where $r(\cdot, \cdot)$ is the latent score function so-called reward model that generates preference for $(\mathbf{x}, y)$. It is clear that

$$
\mathrm{PM}(P_{\mathbf{X}Y}; P_\theta) = \mathrm{RM}(P_{\mathbf{X}Y}; P_\theta) - \log(1 - e^{\mathrm{RM}(P_{\mathbf{X}Y};P_\theta)}),
$$
$$
\mathrm{RM}(P_{\mathbf{X}Y}; P_\theta) = \mathrm{PM}(P_{\mathbf{X}Y}; P_\theta) - \log(1 + e^{\mathrm{PM}(P_{\mathbf{X}Y};P_\theta)}).
$$

Therefore, $\mathrm{PM}(P_{\mathbf{X}Y}; P_\theta)$ and $\mathrm{RM}(P_{\mathbf{X}Y}; P_\theta)$ exhibit a mutual equivalence, i.e., *increase in* $PM(P_{\mathbf{X}Y}; P_\theta)$ *corresponds to the increase in* $RM(P_{\mathbf{X}Y}; P_\theta)$*, and vice versa.* In Lemma 4.3, we establish an upper bound for the absolute difference between EMI and $\mathrm{PM}(P_{\mathbf{X}Y}; P_\theta)$, thereby demonstrating their closeness and ultimately highlighting a connection between EMI and LLM-judge score, e.g., Eq (2).

**Lemma 4.3.** *Given a distribution $P_{\mathbf{X}Y}$ and an MLLM $P_\theta$, if $\mathbb{E}_{\mathbf{x} \sim P_{\mathbf{X}}}[D_{\mathrm{KL}}(P_\theta(\cdot|\mathbf{x}) \| P_{Y|\mathbf{X}=\mathbf{x}})] \leq \delta$, and let the reward function $r(\mathbf{x}, y)$ be $\log P_{Y|\mathbf{X}=\mathbf{x}}(y)$, then*

$$
|EMI(P_{\mathbf{X}Y}; P_\theta) - PM(P_{\mathbf{X}Y}; P_\theta)| \leq \delta + 4.4\delta^{\frac{1}{8}}.
$$

Intuitively, Lemma 4.3 shows that if MLLM $P_\theta$ can approximate the given distribution $P_{\mathbf{X}Y}$ with approximate error $\delta$, the difference between EMI and PM can be bounded by a small term w.r.t. the approximate error $\delta$. Furthermore, assuming that the model class $\{P_\theta : \forall \theta \in \Theta\}$ has sufficient expressive power (i.e., Eq. (8)), we can derive an additional bound for the case of the optimal solution of autoregressive objective (i.e., Eq. (1)), as shown below.

**Theorem 4.4.** *Given a distribution $P_{\mathbf{X}Y}$ with $P_{\mathbf{X}Y} > c > 0$ for some constant or $P_{Y|\mathbf{X}} > c$, if the $\epsilon$-representation capacity assumption holds, i.e.,*

$$
\min_{\theta \in \Theta} \mathbb{E}_{\mathbf{x} \sim P_{\mathbf{X}}}[D_{\mathrm{KL}}(P_{Y|\mathbf{X}=\mathbf{x}} \| P_\theta(\cdot|\mathbf{x}))] \leq \epsilon, \quad (8)
$$

*and let the reward function $r(\mathbf{x}, y)$ be $\log P_{Y|\mathbf{X}=\mathbf{x}}(y)$, then*

$$
|EMI(P_{\mathbf{X}Y}; P_\theta^*) - PM(P_{\mathbf{X}Y}; P_\theta^*)| \leq \delta + 4.4\delta^{\frac{1}{8}},
$$

*where $\theta^*$ is the optimal solution of Eq. (1) over $P_{\mathbf{X}Y}$, and $\delta = 4.4\epsilon^{\frac{1}{8}} - \log c\sqrt{2\epsilon}$.*

Theorem 4.4 shows that with a sufficiently expressive model class, EMI exhibits a stronger alignment with PM when the optimal MLLM parameter $\theta^*$ is obtained through the autoregressive objective. This alignment underscores the validity of using EMI as a reliable metric for evaluating MLLM and quantifying the relative preference of responses.

Although we confine our analysis to $I(P_{\mathbf{X}Y})$ and $\mathrm{EMI}(P_{\mathbf{X}Y}; P_\theta)$, the chain rule of MI, i.e., $I(P_{\mathbf{X}Y}) = I(P_{X_vY}) + I(P_{X_tY|X_v})$ allows us to further factorize the query-response relevance into two input modalities, which is suitable for multimodal LLM's fine-grained evaluation.

### 4.3. Characterizing MLLM Performance Gap via Effective Mutual Information Difference

Now, based on EMI, we are ready to establish formal guarantees on the performance gap of MLLM via *effective mutual information difference* (EMID). EMID is defined as the difference between the EMI on the ID distribution $P_{\mathbf{X}Y}$ and the OOD distribution $Q_{\mathbf{X}Y}$, as follows:

$$
\mathrm{EMID}(P_{\mathbf{X}Y}, Q_{\mathbf{X}Y}; P_\theta) := \mathrm{EMI}(P_{\mathbf{X}Y}; P_\theta) - \mathrm{EMI}(Q_{\mathbf{X}Y}; P_\theta). \tag{9}
$$

To elucidate the key insight and provide a clear foundation, we begin by analyzing a simple scenario where the conditional variables remain consistent across both ID and OOD distributions. In this case, we can derive an upper bound for EMID, as stated in Theorem 4.5. This bound enables us to characterize the maximum performance gap of MLLM over two distributions by measuring the severity of the marginal distribution shift over visual and language modalities.

**Theorem 4.5** (Simplified Scenario). *Given an MLLM $P_\theta$ and distributions $P_{\mathbf{X}Y}$, $Q_{\mathbf{X}Y}$ which have consistent conditional distributions over variables $X_v|X_t$, $X_t|X_v$, and $Y|\mathbf{X}$, if there exist some constants $\delta_P$ and $\delta_Q$ such that*

$$D_{\mathrm{JS}}(P_{Y_\theta}\|P_Y) \le \delta_P, \quad D_{\mathrm{JS}}(Q_{Y_\theta}\|Q_Y) \le \delta_Q, \quad \Delta = \delta_P + \delta_Q$$

*and denote $P_{Y_\theta} = \mathbb{E}_{P_{\mathbf{X}}}[P_\theta(\cdot|\mathbf{x})]$ and $Q_{Y_\theta} = \mathbb{E}_{Q_{\mathbf{X}}}[P_\theta(\cdot|\mathbf{x})]$, then EMID$(P_{\mathbf{X}Y}, Q_{\mathbf{X}Y}; P_\theta)$ is upper bounded by*

$$\widehat{H}\big(D_{\mathrm{JS}}^{\frac{1}{2}}(P_{X_v}\|Q_{X_v}) + D_{\mathrm{JS}}^{\frac{1}{2}}(P_{X_t}\|Q_{X_t})\big) + 8\Delta^{\frac{1}{4}}, \quad (10)$$

*where $\widehat{H} = \max_{\mathbf{x}\in\mathcal{X}}[H(Q_{Y|\mathbf{X}=\mathbf{x}}) + H(P_\theta(\cdot|\mathbf{x}))]$.*

> **Implication.** Theorem 4.5 implies that in the simplified scenario, EMID depends on two main factors: (1) the divergence between the marginal distributions of the visual and textual inputs; (2) the divergence between the model's predictions and the true output distributions, encapsulated by $\delta_P$ and $\delta_Q$.

Theorem 4.5 naturally captures special cases such as visual-only or text-only input shifts. For a visual-only shift, where $D_{\mathrm{JS}}(P_{X_t}\|Q_{X_t}) = 0$, the EMID upper bound depends primarily on the divergence between visual input distributions, and output discrepancy terms. Similarly, for a text-only shift, the bound reflects the divergence in textual input distributions, and output discrepancy terms. These cases not only underscore the flexibility of Theorem 4.5 in isolating the impact of modality-specific distribution shifts on model performance but also highlight the importance of visual and text input shifts on it. In Appendix D, we provide a looser yet better interpretable version of this upper bound (Corollary D.12) by replacing $\Delta$ with discrepancy terms between model output and ground truth conditional distributions.

**General scenario.** Moving beyond the simplified scenario, we now consider the general scenario in which no assumptions are made about the consistency of conditional distributions across ID and OOD settings. This more realistic scenario accommodates shifts not only in the marginal distributions of visual and textual inputs but also in their conditional dependencies and the relationships between inputs and outputs. By relaxing these constraints, we aim to capture the full complexity of distributional shifts encountered

in practice and analyze how such shifts collectively influence the performance gap of MLLMs. The formal upper bound is provided in Theorem 4.6.

**Theorem 4.6** (General Scenario). *Given $P_{\mathbf{X}Y}$ and $Q_{\mathbf{X}Y}$ distributions and an MLLM $P_\theta$, if there exist some constants $\delta_P$ and $\delta_Q$ such that*

$$D_{\mathrm{JS}}(P_{Y_\theta}\|P_Y) \le \delta_P, \quad D_{\mathrm{JS}}(Q_{Y_\theta}\|Q_Y) \le \delta_Q, \quad \Delta = \delta_P + \delta_Q$$

*and denote $P_{Y_\theta} = \mathbb{E}_{P_{\mathbf{X}}}[P_\theta(\cdot|\mathbf{x})]$ and $Q_{Y_\theta} = \mathbb{E}_{Q_{\mathbf{X}}}[P_\theta(\cdot|\mathbf{x})]$, then EMID$(P_{\mathbf{X}Y}, Q_{\mathbf{X}Y}; P_\theta)$ is upper bounded by*

$$\widehat{H}\big(D_{\mathrm{JS}}^{\frac{1}{2}}(P_{X_v}\|Q_{X_v}) + D_{\mathrm{JS}}^{\frac{1}{2}}(P_{X_t}\|Q_{X_t})\big)$$
$$+\widehat{H}\big(\bar{D}_{\mathrm{JS}}^{\frac{1}{2}}(P_{X_t|X_v}\|Q_{X_t|X_v}) + \bar{D}_{\mathrm{JS}}^{\frac{1}{2}}(P_{X_v|X_t}\|Q_{X_v|X_t})\big)$$
$$+4\mathbb{E}_{\mathbf{x}\sim P_{\mathbf{X}}}D_{\mathrm{JS}}^{\frac{1}{4}}(P_{Y|\mathbf{X}=\mathbf{x}}\|Q_{Y|\mathbf{X}=\mathbf{x}}) + 8\Delta^{\frac{1}{4}},$$

*where $\widehat{H} = \max_{\mathbf{x}\in\mathcal{X}}[H(Q_{Y|\mathbf{X}=\mathbf{x}}) + H(P_\theta(\cdot|\mathbf{x}))]$ and*

$$\bar{D}_{\mathrm{JS}}(P_{X|X'}\|Q_{X|X'}) := \mathbb{E}_{\mathbf{x}\sim P_{X'}}[D_{\mathrm{JS}}(P_{X|X'=\mathbf{x}}\|Q_{X|X'=\mathbf{x}})]$$
$$+ \mathbb{E}_{\mathbf{x}\sim Q_{X'}}[D_{\mathrm{JS}}(P_{X|X'=\mathbf{x}}\|Q_{X|X'=\mathbf{x}})].$$

> **Implication.** Compared to Theorem 4.5, Theorem 4.6 indicates that, in the general case, EMID is also influenced by divergences in conditional distributions. Specifically, EMID is upper bounded by marginal distribution shifts in visual and textual inputs ($X_v$ and $X_t$); divergence between marginal output and model response distributions; shifts in conditional dependencies ($X_v|X_t$ and $X_t|X_v$); and a shift between conditional output distributions ($Y|\mathbf{X}$).

Theorem 4.6 holds for broader cases, whereas Theorem 4.5 is much simpler to analyze. Thus, we focus on the validation of Theorem 4.5 in the following section. If we have some knowledge of the data-generating process of $P_{\mathbf{X}Y}$ and $Q_{\mathbf{X}Y}$, we can choose the one that is suitable for given distributions. Although the bounds in both theorems are not such tight, they help us to understand the source of the MLLM performance gap. That is, both Theorem 4.5 and 4.6 *provide an analytic tool to characterize the performance gap of MLLM, representing the first formal framework for evaluating MLLM under distribution shifts.*

## 5. Empirical Validation on Real Benchmark

**Setup.** As done in the pilot experiments (Figure 1 and 6), we mainly used LLaVA v1.5 (Liu et al., 2024a) and LLaVA NeXT (Liu et al., 2024b) in 7B and 13B sizes and evaluated them on the LLaVA-Bench COCO and LLaVA-Bench Wild (Liu et al., 2023) datasets to assess open-ended generation quality. We also considered two advanced MLLMs: Qwen2.5-VL-7B (Bai et al., 2025)

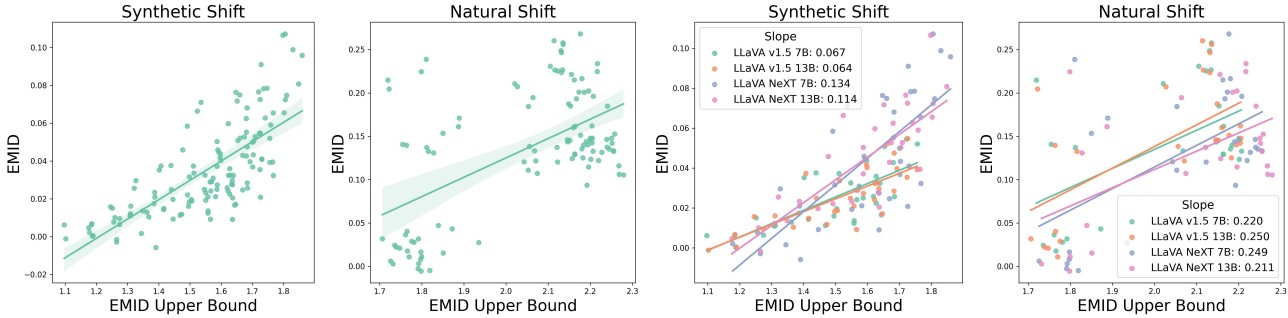

*Figure 3.* **Scatter plot with regression line between empirical estimates of EMID and its upper bound.** Over the 34 synthetic and 27 natural distribution shift scenarios, we evaluate four MLLMs and get 136 cases and 108 cases of synthetic shifts and natural shifts, respectively, for visualizing EMID and its scale-adjusted upper bound estimates (See Appendix B for details). The two panels on the left show results for all four models, whereas the right ones distinguish them per model with fitted linear regression coefficients (Slope).

*Table 1.* **Summary of distribution shift scenarios.**

| Type | Strategy (# of category) |
|---|---|
| Synthetic visual shift (COCO Images) | Perturbation (2): Defocus blur, frost Severity (3): Weak, Normal, Strong |
| Synthetic text shift | Perturbation (2): Typo, Word Replacement Severity (2): Weak, Strong |
| Natural visual shift (Wild Images) | LLaVA-Bench Wild *or* LLaVA-Med Split (3): Easy, Normal, Hard |
| Natural text shift | Translation (6): GE, CH, KO, EL, AR, HI |

and InternVL2.5-7B (Chen et al., 2024), and a domain-specialized open-ended question answering dataset, LLaVA-Med (Li et al., 2024), to explore the broad applicability of our framework. For a comprehensive examination on diverse types of shifts, we further simulate synthetic distribution shifts as well as natural distribution shifts. For synthetic shifts, we consider 7 visual scenarios (1 ID case + 2 synthetic perturbation types at 3 severity levels), and 5 text scenarios (1 ID case + 2 synthetic perturbation types at 2 severity levels), resulting in $7 \times 5 = 35$ synthetic scenarios, where 1 scenario is ID and the other 34 are OOD cases. For natural shifts, we use 4 visual scenarios (1 ID + 3 OOD difficulty levels) and 7 text scenarios (1 ID-English + 6 different languages), yielding $4 \times 7 = 28$ natural scenarios. **This comprehensive design covers 34 synthetic and 27 natural shifts spanning 61 shift scenarios in total**. A summary of the OOD construction strategies is in Table 1.

**Estimation of MI and JSD.** For the empirical realization of our theoretical statements, we adopt a popular neural estimator for MI, CLUB (Cheng et al., 2020) to compute empirical EMI and EMID, and a JS divergence estimator, RJSD (Hoyos-Osorio & Sanchez-Giraldo, 2023), to compute empirical estimates of EMID upper bound on top of embeddings of $\mathbf{X}$, $Y$, and $\hat{Y}$, extracted from CLIP-ViT-B/32 (Radford et al., 2021) and XLM-RoBERTa-base (Conneau, 2019) (see Appendix B for details). Experiments with

*Table 2.* **Spearman rank correlation and Kendall's tau between relative preference (RP) score and EMI.** We conduct correlation analysis between the RP score (Eq. (2)) and EMI (Eq. (7)) on 34 synthetic and 27 natural distribution shifts across four MLLMs.

| | Model | Spearman | | Kendall | |
|---|---|---|---|---|---|
| | | $\rho$ | $p$-val | $\tau$ | $p$-val |
| Synthetic | LLaVA v1.5 7B | 0.794 | <0.001 | 0.604 | <0.001 |
| | LLaVA v1.5 13B | 0.652 | <0.001 | 0.483 | <0.001 |
| | LLaVA NeXT 7B | 0.738 | <0.001 | 0.564 | <0.001 |
| | LLaVA NeXT 13B | 0.726 | <0.001 | 0.527 | <0.001 |
| Natural | LLaVA v1.5 7B | 0.610 | 0.001 | 0.450 | 0.001 |
| | LLaVA v1.5 13B | 0.720 | <0.001 | 0.575 | <0.001 |
| | LLaVA NeXT 7B | 0.593 | 0.001 | 0.435 | 0.001 |
| | LLaVA NeXT 13B | 0.457 | 0.014 | 0.321 | 0.017 |

23 alternative implementations with other estimators derived consistent conclusions (Please refer to Table 7 and 8).

**Correlation between RP score and EMI.** We first conduct the Spearman correlation analysis and Kendall's tau analysis between the RP score and our empirical estimates of EMI. In Table 2, we can see that EMI estimates exhibit a strong correlation with RP score, both in terms of absolute coefficient and $p$-value, across all models. This empirical evidence validates the theoretical connection between EMI and RP score discussed in Theorem 4.4. Therefore, our EMI can be used as a reliable and cost-effective alternative to RP for MLLM evaluation with theoretical grounding.

*Table 3.* **Pearson correlation analysis between EMID and its upper bound.** We provide Pearson $r$ and $p$-value between the empirical estimates of EMID (Eq. (9)) and its upper bound (Eq. (10)) on 34 synthetic and 27 natural shift scenarios per model.

| Model | Synthetic | | Natural | |
|---|---|---|---|---|
| | Pearson $r$ | $p$-val | Pearson $r$ | $p$-val |
| LLaVA v1.5 7B | 0.755 | <0.001 | 0.553 | 0.003 |
| LLaVA v1.5 13B | 0.785 | <0.001 | 0.638 | <0.001 |
| LLaVA NeXT 7B | 0.742 | <0.001 | 0.594 | 0.001 |
| LLaVA NeXT 13B | 0.807 | <0.001 | 0.550 | 0.003 |
| All models | 0.746 | <0.001 | 0.565 | <0.001 |

**Verification of bound.** We now validate our main Theorem. Figure 3 (left two) shows the scatter plots comparing EMID with its upper bound across four models, each evaluated over 34 and 27 synthetic and natural distribution shifts. We see a clear trend between EMID and its upper bound in the synthetic shift where we could directly control the severity of shifts. While the result of the natural shift case is noisier, the overall trend is similar. Meanwhile, our bounds depend on the distributional discrepancy between the model's response $\hat{y}$ and the ground truth response $y$. Thus, they naturally induce different bounds per MLLM. The right panel of Figure 3 presents model-wise plots with linear regression coefficients, where we observe that each model has a different degree of sensitivity against shifts. Pearson correlation analysis results in Table 3 further confirm statistically significant correlations between EMID and its upper bound, supporting the validity of our theorems.

**Partial bound analysis.** It is common that we can not access the ground truth response $Y$ from our training and evaluation datasets. Then, one may want to use a part of EMID upper bound (e.g., JSD terms on visual and textual input only) as an estimator of EMID, by neglecting the output-related term $\Delta$. In Figure 4, we investigate whether the summation of two JS divergence terms can still be predictive for EMID. Although the trends become loose compared to the full bound due to the non-optimality of MLLM parameters, the partial upper bound still has moderately high correlations (denoted by Pearson $r$) with EMID.

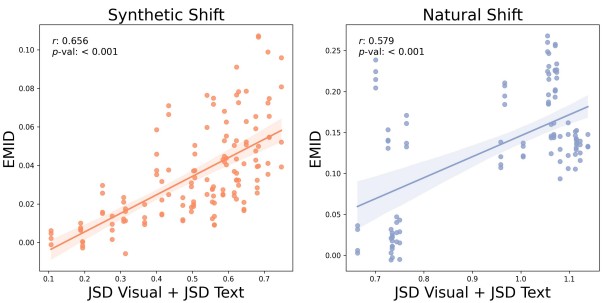

*Figure 4.* **Scatter plot with regression line between empirical estimates of EMID and partial components of its upper bound.** We remove the $\Delta$ term of bound (Eq. (10)) and only use the estimates of JSD terms over visual and text inputs.

**Validation with advanced MLLMs.** So far, our evaluation has focused on LLaVA series MLLMs. Now we validate EMI and EMID upper bound with two state-of-the-art MLLMs, Qwen2.5-VL-7B (Bai et al., 2025) and InternVL2.5-7B (Chen et al., 2024), under synthetic shift scenarios. In Table 4, we see that Qwen2.5-VL model shows strong correlations between EMI and RP score, and between EMID and its upper bound with remarkable statistical significance. Although the correlations between EMI and RP

*Table 4.* **Verification of EMI and EMID bound with advanced MLLMs.** We conduct correlation analysis between EMI and RP, and between EMID and its upper bound (UB) with Qwen2.5-VL-7B (Bai et al., 2025) and InternVL2.5-7B (Chen et al., 2024), and observe statistically significant correlations.

| Model | EMI ⇔ RP | | EMID ⇔ UB |
|---|---|---|---|
| | Spearman $\rho$ ($p$-val) | Kendall $\tau$ ($p$-val) | Pearson $r$ ($p$-val) |
| Qwen2.5-VL | 0.767 (<0.001) | 0.571 (<0.001) | 0.672 (<0.001) |
| InternVL2.5 | 0.375 (0.029) | 0.273 (0.023) | 0.810 (<0.001) |

score are somewhat weak in the case of InternVL2.5, those are still non-negligible correlations (Schober et al., 2018) with the significance level of 0.05, implying the generality of our framework across different models.

**EMI analysis on a specialized domain.** Since the proposed information-theoretic measures rely on embeddings from external models, e.g., CLIP and RoBERTa, to easily compute empirical estimates of MI and JSD, one may wonder whether our framework could be applied to a kind of specialized domain datasets such as medical visual question answering. To investigate this, we adopt LLaVA-Med instruction dataset (Li et al., 2024) as a domain-specific open-ended benchmark on the medical images and corresponding questions. We demonstrate, in Table 5, that EMI and EMID upper bound show strong correlation with RP score and EMID, respectively, indicating the broad applicability of our method in a specialized domain as well.

*Table 5.* **Verification of EMI and EMID bound on LLaVA-Med dataset.** We conduct correlation analysis between EMI and RP, and between EMID and its upper bound (UB) with LLaVA v1.5 (7B) on open-ended medical domain QA tasks.

| EMI ⇔ RP | | EMID ⇔ UB |
|---|---|---|
| Spearman $\rho$ ($p$-val) | Kendall $\tau$ ($p$-val) | Pearson $r$ ($p$-val) |
| 0.718 (<0.001) | 0.572 (<0.001) | 0.930 (<0.001) |

**Application: EMID upper bound as a regularization.** Although our framework mainly stands for the performance analysis of MLLMs under distribution shifts at inference time, one can also leverage EMI and/or EMID upper bound at training time. We give a simple example that instantiates the EMID upper bound as an additional regularization term during visual instruction tuning as below,

$$\mathbb{E}[H(P_\theta(\cdot|\mathbf{x}))] \cdot (D_{\mathrm{JS}}^{\frac{1}{2}}(P_{Z_v}||\mathcal{N}) + D_{\mathrm{JS}}^{\frac{1}{2}}(P_{Z_t}||\mathcal{N}), \quad (11)$$

where $\mathbf{Z} = (Z_v, Z_t)$ denotes an intermediate representation of MLLM given $\mathbf{X}$ and $\mathcal{N}$ is an isotropic Gaussian distribution that has the same dimensionality as $Z_v$ and $Z_t$. Here, we utilize a non-informative prior distribution $\mathcal{N}$ as an alternative to the target distribution $Q$ (which is usually inaccessible during training) to implicitly penalize representation discrepancy between $P$ and $Q$. The above term

*Table 6.* **Visual instruction tuning with EMID upper bound.** We use EMID upper bound as a regularization term (Eq. 11) during instruction tuning of LLaVA v1.5 (7B) on a 10% subset of LLaVA-mix-665K w/ and w/o $R$, and report relative preference scores.

| Method | ID | V Shift | T Shift | J Shift |
|---|---|---|---|---|
| Baseline | 72.7 | 65.8 | 68.0 | 59.6 |
| Baseline w/ $R$ | 72.7 | **66.3** | **68.3** | **60.8** |

encourages intermediate representations of visual and text inputs to be shrunk into a zero-centered Gaussian where the penalizing strength is scaled by an averaged model output entropy across batch samples. Table 6 presents evaluation results of LLaVA v1.5 7B model on LLaVA-Bench COCO (ID) and its synthetic shift variants (V, T, and J Shift) after being instruction tuned on LLaVA-mix-665k subset, where we see our EMID-based regularization effectively improves robustness to shifts while maintaining the ID performance.

# 6. Related Works

**Fine-tuned foundation models under distribution shifts.** Recent findings imply that fine-tuning on a relatively small amount of ID datasets hurts the OOD generalization capability of foundation models (Kumar et al., 2022; Wortsman et al., 2022). Although lots of follow-up studies (Goyal et al., 2023; Tian et al., 2023; Oh et al., 2025) including theory-inspired methods (Kumar et al., 2022; Ju et al., 2022; Oh et al., 2024) have been proposed, almost all of them focused on a discriminative model such as CLIP (Radford et al., 2021) for the image classification task. Given the rapidly growing popularity of MLLMs, it is necessary to investigate the reliability of MLLMs under distribution shifts with a tangible formulation. We lay a cornerstone for this.

**Performance analysis of MLLM.** There have been numerous reports on MLLMs' corner-case behaviors. Zhang et al. (2024a), Zhou et al. (2024), and Verma et al. (2024) observed that MLLMs poorly perform under specialized domains or synthetic perturbation, while Zhai et al. (2024) and Zhang et al. (2024b) showed that MLLMs are bad at some simple image classification tasks. Besides, Li et al. (2023b) and Ye-Bin et al. (2025) focused on the object hallucination of MLLM under spurious correlation. However, they all lacked a formal framework to explain such degradation of MLLMs. We recast the degeneration of MLLMs via robustness under distribution shifts between instruction-tuning and evaluation data (Liang et al., 2025), and devise the first theoretical framework to analyze MLLMs.

**Information-theoretic approach for model evaluation.** The information-theoretic view has been steadily adopted to establish evaluation criteria for language model probing (Hewitt et al., 2021), prompt engineering (Sorensen et al., 2022), and rationale evaluation (Chen et al., 2023), alongside learning objectives (Alemi et al., 2016; Chen et al., 2016; Tschannen et al., 2020; Kong et al., 2020; Wang et al., 2021), but relatively unexplored for MLLM yet. We also note some works adopting information-theoretic approaches to analyze models under distribution shifts (Federici et al., 2021; Shui et al., 2022). Although they focused on classification tasks with discriminative models, we established new theorems for MLLM analysis based on a new metric, EMI.

# 7. Discussion

This work urged the development of a formal framework to understand MLLMs under distribution shifts which is unexplored yet crucial for reliable artificial intelligence in the wild. As a first step for this, we devised effective mutual information (EMI) as a metric for MLLM evaluation and showed its theoretical connection to an existing standard metric, relative preference score. Then, we provide a theoretical upper bound for an MLLM's EMI difference between ID and OOD that consists of JS divergences for marginal and conditional distributions of input/output variables. Through experiments on benchmarks spanning 61 distribution shifts, we show the correlation between relative preference and EMI, and further show correlations between EMI difference and its upper bound, thereby empirically verifying our theoretical claims across various models.

**Practical implication.** As shown in Table 2, EMI strongly correlates with RP. Compared to RP, the MI estimator can be computed more efficiently without relying on the computationally expensive judge LLM (Achiam et al., 2023) (see Appendix C.4). Thus, EMI can be used as a theoretically grounded, cost-effective evaluation metric that measures the effective relevance between multimodal queries and open-ended responses. Besides, the upper bound of EMID can be adopted as a regularizer during model training, as we showed in Table 6, or test-time adaptation of MLLM to improve its robustness to distribution shifts (Li et al., 2023a), as well as a robustness measure of MLLM.

**Limitation and future work.** Although input-output relevance measured by EMI is one of the most important properties for instruction-following models, other crucial quality attributes are not captured by the form of the relevance term. Extending the theory to support evaluation across multiple facets of MLLM will be worth exploring. In addition, we only simulated some intuitive types of distribution shifts with a simplified assumption for the data structure, leaving the analysis on some complex shifts driven by spurious correlation (Simon, 1954) that may be covered by Thm. 4.6. Meanwhile, despite its consistent correlation to EMID, our upper bound is not quite tight in theory, and we did not discuss a lower bound of EMID. Pursuing a tighter bound or exploring the lower limit of EMID can also be worthwhile.

## Acknowledgement

We gratefully appreciate the ICML anonymous reviewers for their valuable feedback, and also appreciate constructive comments from Max Khanov, Min-Hsuan Yeh, Dongkwan Kim, and Kyungeun Lee. Changdae Oh, Shawn Im, Xuefeng Du, and Yixuan Li are supported by the AFOSR Young Investigator Program under award number FA9550-23-1-0184, National Science Foundation (NSF) Award No. IIS-2237037 and IIS-2331669, Office of Naval Research under grant number N00014-23-1-2643, Schmidt Sciences Foundation, and Alfred P. Sloan Fellowship. Also, Zhen Fang was funded by the Australian Government through the Australian Research Council (ARC) under grant number DE250100363.

## Impact Statement

This work lays the first theoretical foundation for quantifying the reliability of MLLMs. The theoretical statements we made shine a light on analyzing the MLLM performance gap under distribution shifts, which commonly emerged in real-world applications. Our framework can help aware of the potential risk, i.e., performance variation, of MLLMs, and guide practitioners to devise a method towards robust adaptation for chat assistants, thereby ensuring the trustworthiness of artificial intelligence solutions in crucial social applications such as finance and healthcare.

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

# Appendix

## Contents

## A. Detailed Description for Effective Mutual Information

We propose effective mutual information (EMI) as an alternative to vanilla mutual information (MI) to evaluate a model-generated output response given an input query. As explained in Section 4.2, MI (e.g., $I(P_\mathbf{X} \otimes P_\theta)$) can not take into account the intrinsic characteristics of data distribution. See Figure 5 for an intuitive example. The amount of information represented by entropy $H(\cdot)$ and conditional entropy $H(\cdot|\cdot)$ can vary depending on the data-generating process of each dataset. For example, if the task we are interested in is closer to solving a narrow problem in some specific domain (e.g., OOD1: LLaVA-Med; Li et al. (2024)), the cardinality of the desired output response space may be significantly smaller than that of a general problem-solving task in a general domain (e.g., OOD2: LLaVA-Bench Wild; Liu et al. (2023)), and the ground truth MI can differ depending on the domain. By considering these baseline amounts of information, EMI can measure how much our model captures **effective relevance** between input and output.

In Section 4.2, we provide some justifications for using EMI as an evaluation metric of MLLMs by revealing analogies to excess risk and effective robustness and presenting its theoretical connection to relative preference score. While LLM-as-a-Judge enables flexible evaluation for open-ended generation tasks with multiple user-defined criteria, EMI confines the facet of evaluation to query-response relevance. However, compromise in the flexibility of evaluation enables us to build solid theoretical statements that are necessary for understanding MLLMs to shifts and improving them in a principled way.

Meanwhile, as we adopt neural network models for empirical estimation of EMI, it is somewhat similar to the model-based heuristic metrics, such as BERTscore (Zhang et al., 2020), BARTscore (Yuan et al., 2021), and CLIPscore (Hessel et al., 2021), that map input(s) to a scalar score through a single forward evaluation of the model. However, we take a step further beyond the simple working-heuristic method and lay a theoretical foundation with EMI.

## B. Implementation Details

In this paper, we proposed EMI for a reliable evaluation of multimodal large language models (MLLMs) with a theoretical ground. Based on EMI, to analyze the MLLM performance gap under distribution shift, we provided the upper bound for EMI difference between ID and OOD data. In this section, we describe the procedures for estimating EMI and its upper

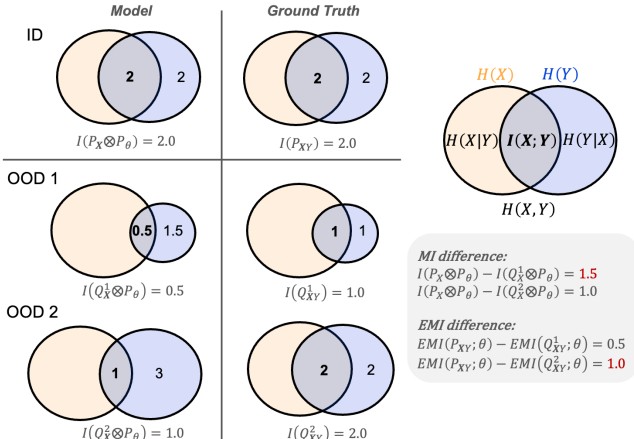

*Figure 5.* **Information diagram and motivation of effective mutual information.** The difference between vanilla MI terms does not consider the domain-dependent intrinsic scale and mutual information, thereby failing to fairly measure the relevance between input query $x$ and model prediction $\hat{y}$. Meanwhile, EMI ablates the domain-dependent characteristic to focus on measuring effective relevance between $x$ and $\hat{y}$.

bound in detail.

**Overview.** To estimate EMI and its upper bound, we first need to define estimators for MI and Jensen-Shannon divergence (JSD). Those estimators commonly adopt neural network encoders to project the raw data such as text and image into embedding space of neural networks to reduce problem complexity (Oord et al., 2018; Liu et al., 2020), and then, MI estimator commonly optimizes a simple critic function (Poole et al., 2019) on top of the embeddings of data. After training of MI estimator, we evaluate empirical MI over different data distributions. For JSD estimation, given the embedding spaces of pre-trained models, additional training is not necessary. Therefore, the procedures can be divided into two phases: (1) neural MI estimator training, and (2) inference of MI and JSD.

**MI estimation.** Estimating MI with finite samples from unknown population distribution is a non-trivial problem, and has been actively studied (Fraser & Swinney, 1986; Paninski, 2003; Kraskov et al., 2004; Nguyen et al., 2010; Shwartz-Ziv & Tishby, 2017; Belghazi et al., 2018; Poole et al., 2019; Cheng et al., 2020). We adopted the contrastive log-ratio upper bound (CLUB; Cheng et al. (2020)) as our default MI estimator similar to (Cheng et al., 2021). We first extract embeddings for visual input query $Z_v = \text{enc}_v(X_v)$ and text input query $Z_t = \text{enc}_t(X_t)$ from visual and text encoder models and take the mean of them to provide input query embedding $Z_{\mathbf{X}} = \frac{Z_v + Z_t}{2}$. Specifically, we adopt the most representative embedding models for each modality, i.e., CLIP pre-trained[1] ViT-B/32 and XLM-RoBERTa-Base[2] (Conneau, 2019) as visual and text encoders, respectively by default. We also obtain the embedding vectors for the model response $Z_{\hat{Y}} = \text{enc}_t(\hat{Y})$ and reference response $Z_Y = \text{enc}_t(Y)$ with text encoder model. Then, we train the MI estimator $\hat{I}_\psi(\cdot, \cdot)$ with parameter $\psi$ via gradient descent. To be specific, CLUB formulates the unbiased estimation for MI as below,

$$\hat{I}_{\text{CLUB}}(P_{Z_{\mathbf{X}} Z_Y}) = \frac{1}{N} \sum_{i=1}^{N} \log q_\psi(z_{y_i} | z_{x_i}) - \frac{1}{N^2} \sum_{i=1}^{N} \sum_{j=1}^{N} \log q_\psi(z_{y_j} | z_{x_i}), \tag{12}$$

where $q_\psi(\cdot | \cdot)$ denotes variational approximation of ground truth probability density function $p(\cdot | \cdot)$.

Following (Cheng et al., 2020; 2021), we parameterize the $q_\psi$ as a multi-variate Gaussian distribution and estimate the mean and variance parameters of Gaussian with separated two-layer MLPs with 250 hidden dimension size. During mini-batch training, those MLPs consume the concatenated input and response embeddings $\{[z_{\mathbf{x}_i}, z_{y_i}]\}_{i=1}^{N}$ to produce a scalar estimate of MI, and they are simultaneously optimized by AdamW optimizer with learning rate 0.001 and batch size 1,024 for 5,000 iterations. However, if we have to train an estimator for every ID-OOD data pair, it may not be practical when the number

---

[1]https://github.com/openai/CLIP.
[2]https://huggingface.co/FacebookAI/xlm-roberta-base

of data pairs to be evaluated is large. Therefore, we constructed a dataset that integrates all ID-OOD data subsets for MI training (integration of all variants of LLaVA-Bench datasets reach roughly 5,000 samples for natural shift, and 9,000 samples for synthetic shift), trains it only once, and then infers all ID-OOD scenarios (27 for natural shift, 34 for synthetic shift) using these common MI estimators. This not only significantly reduces the time required to evaluate the model's robustness against multiple distribution shift scenarios, but also stabilizes the training process by increasing the size of the data set used in the training process.

**JS divergence estimation.**   Estimation of distribution divergences from finite samples has been also a central topic of research (Yang & Barron, 1999; Sriperumbudur et al., 2012; Li & Turner, 2016; Bu et al., 2018; Sinn & Rawat, 2018; Sreekumar & Goldfeld, 2022; Hoyos-Osorio & Sanchez-Giraldo, 2023). We adopt the most recent one, representation Jensen-Shannon divergence (RJSD; Hoyos-Osorio & Sanchez-Giraldo (2023); Hoyos & Giraldo (2024)), which proves its effectiveness on real benchmark datasets as our JSD estimator. The formula is as follows:

$$\hat{D}_{\text{RJSD}}(P, Q) = S(\frac{C_P + C_Q}{2}) - \frac{1}{2}(S(C_P) + S(C_Q)), \tag{13}$$

where $C_P = \mathbb{E}_{X \sim P}[\phi(X) \otimes \phi(X)]$ and $S(C_P) = -\text{Trace}(C_P \log C_P)$. Similar to MI, we compute $\hat{D}_{\text{RJSD}}(P, Q)$ in the embedding space of the same frozen pre-trained models, i.e., leverage neural network embedding space as a kernel $< \phi(x), \phi(x') >$. In contrast to the case of MI, RJSD with a frozen neural embedding model does not require additional training. Still, one might consider learning the embedding model from scratch if necessary.

**Scale-adjusted EMID upper bound construction.**   By leveraging the estimator described above, one can compute all JSD terms in the proposed EMID upper bound (Eq. 10): $D_{\text{JS}}(P_{X_v} \| Q_{X_v})$, $D_{\text{JS}}(P_{X_t} \| Q_{X_t})$, $D_{\text{JS}}(P_{Y_\theta} \| P_Y)$, and $D_{\text{JS}}(Q_{Y_\theta} \| Q_Y)$). Because the exact computation of the entropy scaler term $\hat{H}$ in Eq. (10) is intractable, we relax it with an estimate on batch samples, and replace the inaccessible true conditional distribution $Q_{Y|\mathbf{X}}$ with $P_\theta$. For a case of sentence output $Y = \{y_1, ..., y_L\}$ with $L$ tokens, the length-normalized batch entropy estimate over $N$ samples indexed through $i \in \mathcal{I} = \{1, ..., N\}$ can be formulated as below,

$$\tilde{H} = \max_{i \in \mathcal{I}} -\frac{1}{L} \sum_{l=1}^{L} \log P_\theta(y_{i,l} | y_{i,<l}, x_i). \tag{14}$$

In pilot experiments, we observed that the values of $\tilde{H}$ are centered on 2.0 in the datasets that we considered in this work. Therefore, $\hat{H} = \max_{\mathbf{x} \in \mathcal{X}}[H(Q_{Y|\mathbf{X}=\mathbf{x}}) + H(P_\theta(\cdot | \mathbf{x}))]$ can approximately have a value 4.0. As the main implications of EMID upper bound are providing the characterization of MLLM performance gaps and serving as a practical measure of robustness, we compute the scale-adjusted upper bound estimates for EMID upper bound (UB) as below,

$$\text{EMID UB} = \hat{H}\big(D_{\text{JS}}^{\frac{1}{2}}(P_{X_v} \| Q_{X_v}) + D_{\text{JS}}^{\frac{1}{2}}(P_{X_t} \| Q_{X_t})\big) + 4\big(D_{\text{JS}}^{\frac{1}{4}}(P_{Y_\theta} \| P_Y) + D_{\text{JS}}^{\frac{1}{4}}(Q_{Y_\theta} \| Q_Y)\big)$$
$$\approx 4\big(D_{\text{JS}}^{\frac{1}{2}}(P_{X_v} \| Q_{X_v}) + D_{\text{JS}}^{\frac{1}{2}}(P_{X_t} \| Q_{X_t}) + D_{\text{JS}}^{\frac{1}{4}}(P_{Y_\theta} \| P_Y) + D_{\text{JS}}^{\frac{1}{4}}(Q_{Y_\theta} \| Q_Y)\big)$$
$$\text{Scale-adjusted } \hat{\text{U}}\text{B} := \hat{D}_{\text{RJSD}}^{\frac{1}{2}}(P_{X_v}, Q_{X_v}) + \hat{D}_{\text{RJSD}}^{\frac{1}{2}}(P_{X_t}, Q_{X_t}) + \hat{D}_{\text{RJSD}}^{\frac{1}{4}}(P_{Y_\theta}, P_Y) + \hat{D}_{\text{RJSD}}^{\frac{1}{4}}(Q_{Y_\theta}, Q_Y).$$

See Section D for the detailed derivation of EMID UB. Note that applying the linear transformation to the target variables of Pearson correlation analysis does not affect the correlation coefficient value, so that the correlation coefficients between the EMID and its scale-adjusted UB has the same values as between the EMID and its original UB, even though it is not the exact estimate of EMID UB.

**MLLM judge and relative preference (RP) score.**   For evaluation of open-ended generation tasks, (M)LLM-as-a-Judge has been adopted as a current de facto standard. Following (Liu et al., 2023; 2024a), we use GPT-4[3] with text-only inference mode (with plain-text form visual cue such as ground truth caption for image) as a judge model and also use the output of the same model as a reference answer for each query. We leverage the prompts provided by the source code of LLaVA[4], and compute the RP score of a model of interest by comparing its output with that of the reference answer.

---

[3]`gpt-4-turbo` with `2024-08-01-preview` API version was mainly adopted
[4]https://github.com/haotian-liu/LLaVA

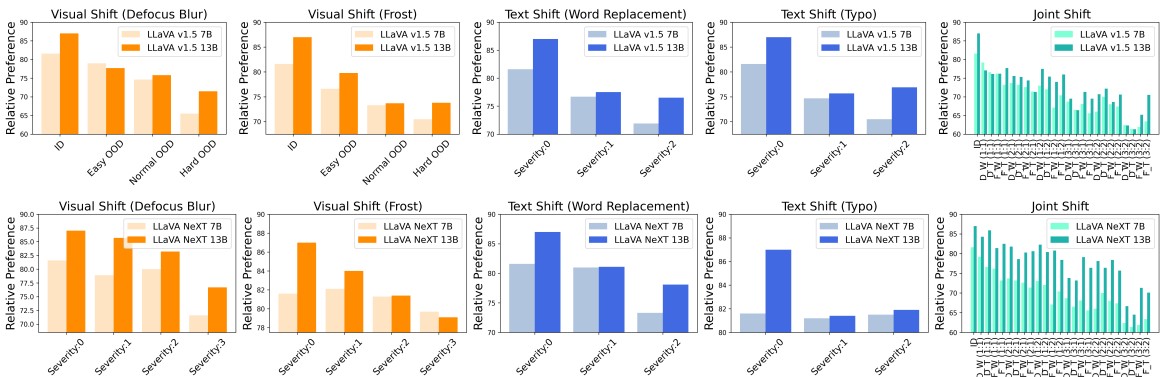

*Figure 6.* **Performance variation against varying degrees of distribution shifts.** We evaluated LLaVA v1.5 and LLaVA NeXT models on 34 out-of-distribution (OOD) variants induced by image and text perturbations of the LLaVA-Bench COCO dataset (ID). Here, the $x$-axis is sorted by the severity of shifts between ID and OOD. There is a consistent trend – increased degrees of distribution shifts result in performance degradations of MLLM.

## C. Extended Empirical Validation and Discussion

### C.1. Additional Result from Pilot Study

In Section 3, we conduct an experiment to validate our hypotheses on the relation between MLLM performance degradation and the severity of natural distribution shift. In Figure 6, we provide additional results from another type of distribution shift that occurred by image and text perturbations. For image perturbation, we consider defocus blur and frost with three different magnitudes, and for text perturbation, we consider keyboard typo error and word synonym replacement with two different magnitudes (We adopt the source code of `https://github.com/Jielin-Qiu/MM_Robustness` to generate perturbed datasets). We observe the consistent trend in the relation between MLLM performance degradation and the severity of distribution shifts for the case of visual-only, text-only, and joint shift, likewise the case of natural shifts in Figure 1. That is, the increased magnitude of distribution shifts induces more severe MLLM performance degradation, and the degree of performance degradation can attribute to shifts in two modalities.

### C.2. Different Design Choices of MI and JSD Estimation

Note that the results of all theorem (Lemma 4.3, Theorem 4.4, Theorem 4.5, and Theorem 4.6) are not limited to a specific class of MI and JSD estimators. To investigate whether our empirical verification of theorems robustly holds in an estimator-agnostic manner (if the estimator is valid), we provide an ablation study for the MI estimator, JSD estimator, and embedding space that the estimators are built on.

Specifically, we consider four MI estimators {NWJ (Nguyen et al., 2010), MINE (Belghazi et al., 2018), InfoNCE (Oord et al., 2018), CLUB (Cheng et al., 2020)}, three embedding spaces {individual models (CLIP ViT and XLM-RoBERTa), E5-V joint (Jiang et al., 2024), E5-V disjoint (Jiang et al., 2024)}, and two JSD estimators {MMD (Liu et al., 2020), RJSD(Hoyos-Osorio & Sanchez-Giraldo, 2023)}. E5-V (Jiang et al., 2024) is a recently proposed embedding extraction method that leverages an MLLM with a carefully designed prompt. We used the default prompt "`Summary above sentence/image in one word:`" to separately extract embeddings (E5-V disjoint) for images and sentences, and design an ensemble of four custom prompts,

1. "`Summary of the image <image>, and sentence <sent> in one word:`"

2. "`Summary of the visual content "<image>" with an associated text query "<sent>" in one word:`"

3. "`Given <image>, Summary of the sentence "<sent>" in one word:`"

4. "`Given visual content "<image>", Summary of the text query "<sent>" in one word:`",

to extract multimodal joint query embedding (E5-V joint) by averaging four embedding vectors per (image, sentence) pair. In Table 7, we conduct Spearman correlation analysis over the 12 (4 × 3) cases of MI estimator and embedding

*Table 7.* **Ablation study for MI estimator and embedding space**. We evaluate four MI estimators with three different embedding space choices in terms of Spearman correlation coefficient $\rho$ between RP score and EMI. We can see that EMI and RP score are robustly correlated to variations in the embedding space and the MI estimator, but CLUB shows the most stable correlation.

| Configuration | | LLaVA v1.5 7B | | LLaVA v1.5 13B | | LLaVA NeXT 7B | | LLaVA NeXT 13B | |
|---|---|---|---|---|---|---|---|---|---|
| MI estimator | Embedding | $\rho$ | $p$-val | $\rho$ | $p$-val | $\rho$ | $p$-val | $\rho$ | $p$-val |
| CLUB | E5-V disjoint | 0.695 | 0.000 | 0.726 | 0.000 | 0.581 | 0.001 | 0.579 | 0.001 |
| CLUB | E5-V joint | 0.910 | 0.000 | 0.846 | 0.000 | 0.817 | 0.000 | 0.902 | 0.000 |
| CLUB | Individual models | 0.606 | 0.001 | 0.720 | 0.000 | 0.594 | 0.001 | 0.457 | 0.014 |
| InfoNCE | E5-V disjoint | 0.670 | 0.000 | 0.708 | 0.000 | 0.638 | 0.000 | 0.590 | 0.001 |
| InfoNCE | E5-V joint | 0.800 | 0.000 | 0.717 | 0.000 | 0.636 | 0.000 | 0.609 | 0.001 |
| InfoNCE | Individual models | 0.519 | 0.005 | 0.421 | 0.026 | 0.410 | 0.030 | 0.275 | 0.157 |
| MINE | E5-V disjoint | 0.664 | 0.000 | 0.605 | 0.001 | 0.269 | 0.167 | 0.278 | 0.153 |
| MINE | E5-V joint | 0.632 | 0.000 | 0.559 | 0.002 | 0.610 | 0.001 | 0.308 | 0.111 |
| MINE | Individual models | 0.632 | 0.000 | 0.562 | 0.002 | 0.632 | 0.000 | 0.613 | 0.001 |
| NWJ | E5-V disjoint | 0.583 | 0.001 | 0.552 | 0.002 | 0.513 | 0.005 | 0.429 | 0.023 |
| NWJ | E5-V joint | 0.502 | 0.005 | 0.519 | 0.005 | 0.492 | 0.008 | 0.480 | 0.010 |
| NWJ | Individual models | 0.510 | 0.006 | 0.717 | 0.000 | 0.488 | 0.008 | 0.322 | 0.095 |

space ablation. We can clearly see that EMI is consistently correlated with the RP score which demonstrates the robust effectiveness of our theorem in practice. Meanwhile, among the candidate MI estimators and embedding space, CLUB and two E5-V joint embeddings show outstanding results. However, E5-V embedding extraction require a forward pass of MLLM in contrast to the case of leveraging relatively small individual models (ViT base and BERT-base). To strike the balance between effectiveness and efficiency, we adopt CLIP ViT-B/32 and XLM-RoBERTa-Base embedding spaces by default.

Next, we present the Pearson correlation analysis result in Table 8 by ablating the JSD estimator with the MI estimator and embedding space choices. Although there are some variations in the exact values, we also observe consistently significant correlations between EMID and its upper bound (that of Theorem 10). Therefore, the upper bound of EMID we derived robustly holds in practice across diverse estimator configurations.

### C.3. Hyperparameter Sensitivity

We provide the hyperparameter configuration for MI estimator training (Table 9) and further provide sensitivity analysis for varying hyperparameters (Figure 7). We see that the CLUB estimator is quite robust to varying hyperparameters, i.e., batch size, learning rate, and hidden dimension, which implies the effectiveness of EMI estimation without intensive hyperparameter tuning.

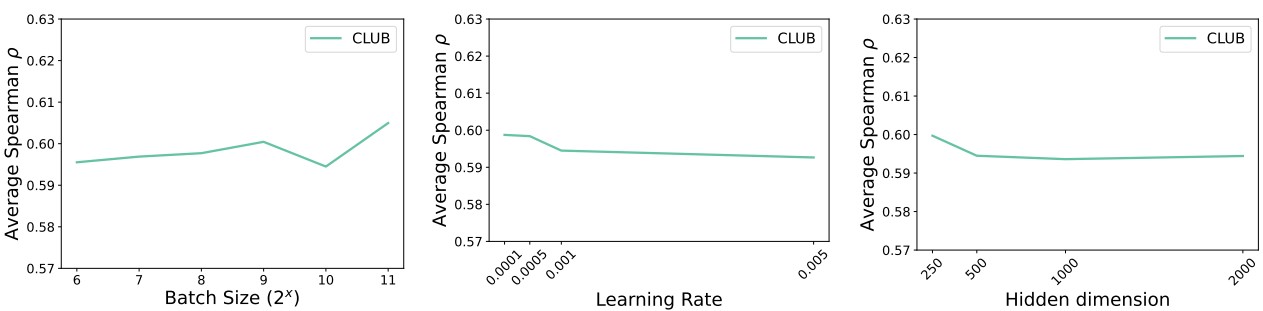

*Figure 7.* **Sensitivity analysis for batch size, learning rate, and hidden dimension during MI estimator training.** For the considered hyperparameter searching grid, MI estimates derived by the CLUB estimator robustly achieve high Spearman correlation with RP score (the largest deviation is less than 0.02 absolute value).

*Table 8.* **Ablation study for MI estimator, JSD estimator, and embedding space**. We evaluate four MI estimator candidates and two JSD estimator candidates, with three different embedding space choices in terms of Pearson correlation coefficient between EMID and its upper bound. In all the considered variations, EMID and the upper bound of EMID (i.e., the simplified version in Theorem 4.5) show strong correlations, implying that our theorem robustly holds in practice.

| Configuration | | | Pearson | |
| --- | --- | --- | --- | --- |
| JSD estimator | MI estimator | Embedding | $r$ | $p$-val |
| RJSD | CLUB | E5-V disjoint | 0.618 | 0.000 |
| RJSD | CLUB | E5-V joint | 0.659 | 0.000 |
| RJSD | CLUB | Individual models | 0.565 | 0.000 |
| RJSD | InfoNCE | E5-V disjoint | 0.618 | 0.000 |
| RJSD | InfoNCE | E5-V joint | 0.617 | 0.000 |
| RJSD | InfoNCE | Individual models | 0.295 | 0.002 |
| RJSD | MINE | E5-V disjoint | 0.602 | 0.000 |
| RJSD | MINE | E5-V joint | 0.534 | 0.000 |
| RJSD | MINE | Individual models | 0.630 | 0.000 |
| RJSD | NWJ | E5-V disjoint | 0.611 | 0.000 |
| RJSD | NWJ | E5-V joint | 0.413 | 0.000 |
| RJSD | NWJ | Individual models | 0.468 | 0.000 |
| MMD | CLUB | E5-V disjoint | 0.618 | 0.000 |
| MMD | CLUB | E5-V joint | 0.659 | 0.000 |
| MMD | CLUB | Individual models | 0.478 | 0.000 |
| MMD | InfoNCE | E5-V disjoint | 0.432 | 0.000 |
| MMD | InfoNCE | E5-V joint | 0.617 | 0.000 |
| MMD | InfoNCE | Individual models | 0.295 | 0.002 |
| MMD | MINE | E5-V disjoint | 0.602 | 0.000 |
| MMD | MINE | E5-V joint | 0.623 | 0.000 |
| MMD | MINE | Individual models | 0.630 | 0.000 |
| MMD | NWJ | E5-V disjoint | 0.611 | 0.000 |
| MMD | NWJ | E5-V joint | 0.273 | 0.004 |
| MMD | NWJ | Individual models | 0.468 | 0.000 |

*Table 9.* **Hyperparameter tuning grid and selected value for MI estimator training.** The hyperparameters are selected based on the variance of last 10 iterations during training.

| Parameter | Selected | Sweep |
| --- | --- | --- |
| learning rate | 0.001 | {0.005, 0.001, 0.0005, 0.0001} |
| batch size | 1024 | {64, 128, 256, 512, 1024, 2048} |
| hidden dimension | 100/500 | {250, 500, 1000, 2000} |

## C.4. Runtime Analysis

In addition to the advantage of allowing rigorous theoretical statements, EMI also has practical advantages over the RP score derived by the LLM judge. Specifically, while both RP score and EMI are model-dependent, the former relies on models with tens to hundreds of billions of parameters, while the latter enables meaningful evaluation even with relatively small embedding models with millions of parameters. To quantitatively argue this, we compare the inference time per instance and for the entire LLaVA-Bench COCO dataset in Table 10. As shown in the table, EMI can shorten the time by 138 times compared to the RP score. Even including the time required for MI training, EMI-based evaluation is still 3 times faster than MLLM judgment-based evaluation. Since MI training is performed only once and then transferred to all ID-OOD scenarios, the efficiency of EMI-based evaluation over MLLM judgment becomes more evident as the number of datasets to be evaluated increases. In addition, in the LLM judge paradigm, although open-source LLM judges (Kim et al., 2024) have been actively studied recently, proprietary LLMs are still dominant in practice, so one must pay per instance query, whereas EMI can make meaningful inferences with publicly available open-source feature extractors without paying per query.

*Table 10.* **Runtime comparison on LLaVA-Bench COCO 90 samples.** We compare the actual wall-clock time (second) of EMI estimation and MLLM judgment protocols by evaluating model-generated response and reference response given input query from the LLaVA-Bench COCO dataset (Liu et al., 2023). EMI estimation protocol on top of CLIP ViT-B/32 and XLM-RoBERTa-Base embedding achieves 138 times boosting from MLLM judgment protocol with GPT-4o.

| | LLaVA v1.5 7B | | LLaVA v1.5 13B | | LLaVA NeXT 7B | | LLaVA NeXT 13B | | Total runtime |
| | per instance | dataset | per instance | dataset | per instance | dataset | per instance | dataset | |
|---|---|---|---|---|---|---|---|---|---|
| MI estimator training | | | | 663.45 | | | | | 663.45 |
| EMI estimation | 0.0388 | 3.5884 | 0.0392 | 3.5652 | 0.0412 | 3.7107 | 0.0411 | 3.7039 | 14.56 (× 138 boosting) |
| MLLM judgment (GPT-4o API) | 5.49 | 493.75 | 5.48 | 493.59 | 5.52 | 496.66 | 5.83 | 524.45 | 2008.45 |

## C.5. Notes on Practical Usage

Although the empirical estimates of EMI and EMID upper bound show consistent correlations with the relative preference score and EMID, the absolute value of MI estimates varies depending on the estimator types and their configurations. Therefore, if one has used the empirical EMI estimates (computed by a two-layer MLP-based CLUB estimator on CLIP-VIT and RoBERTa embedding spaces) to assess the qualities of responses from MLLMs, the same estimator and the same embedding models should be used across targeted models and datasets for fair comparison.

# D. Extended Theoretical Analysis with Full Proof

In this section, we provide proof of all theorems (Lemma 4.3, Theorem 4.4, Theorem 4.5, and Theorem 4.6) in our manuscript, and introduce an additional theoretical result (Corollary D.12).

## D.1. Proof for the relationship between EMI and preference model

First, we provide proof of the closeness between the effective mutual information (EMI) and the preference model.

**Lemma D.1** (Restatement of Lemma 4.3). *Given a distribution $P_{\mathbf{X}Y}$ and an MLLM $P_\theta$, let the reward model function $r(\mathbf{x}, y)$ be $\log P_{Y|\mathbf{X}=\mathbf{x}}(y)$. If $\mathbb{E}_{\mathbf{x} \sim P_{\mathbf{X}}} D_{\mathrm{KL}}(P_\theta(\cdot|\mathbf{x}) \| P_{Y|\mathbf{X}=\mathbf{x}}) \leq \delta$, then,*

$$|EMI(P_{\mathbf{X}Y}; P_\theta) - PM(P_{\mathbf{X}Y}; P_\theta)| \leq \delta + 4.4\delta^{\frac{1}{8}}.$$

*Proof.* Let $P_{Y_\theta} = \mathbb{E}_{\mathbf{x} \sim \mathbf{X}} P_\theta(\cdot|\mathbf{x})$, and note the expression for EMI below,

$$
\begin{aligned}
\mathrm{EMI}(P_{\mathbf{X}Y}; P_\theta) &= I(P_{\mathbf{X}} \otimes P_\theta) - I(P_{\mathbf{X}Y}) \\
&= H(P_{Y_\theta}) - H(P_\theta(\cdot|\mathbf{X})) - H(P_Y) + H(P_{Y|\mathbf{X}}) \\
&= \big(H(P_{Y_\theta}) - H(P_Y)\big) \\
&\quad + \mathbb{E}_{\mathbf{x} \sim P_{\mathbf{X}}}[\mathbb{E}_{\hat{y} \sim P_\theta(\cdot|\mathbf{x})} \log P_\theta(\hat{y}|\mathbf{x})] - \mathbb{E}_{\mathbf{x} \sim P_{\mathbf{X}}}[\mathbb{E}_{y \sim P_{Y|\mathbf{X}=\mathbf{x}}} \log P_{Y|\mathbf{X}=\mathbf{x}}(y)]
\end{aligned}
$$

Next, given $r(\mathbf{x}, y) = \log P_{Y|\mathbf{X}=\mathbf{x}}(y)$, logit Bradley-Terry preference model (PM) (Hunter, 2004) can be expressed as,

$$\mathrm{PM}(P_{\mathbf{X}Y}; P_\theta) = \mathbb{E}_{\mathbf{x} \sim P_{\mathbf{X}}}[\mathbb{E}_{\hat{y} \sim P_\theta(\cdot|\mathbf{x})} \log P_{Y|\mathbf{X}=\mathbf{x}}(\hat{y}) - \mathbb{E}_{y \sim P_{Y|\mathbf{X}=\mathbf{x}}} \log P_{Y|\mathbf{X}=\mathbf{x}}(y)]$$

Therefore,

$$
\begin{aligned}
|\mathrm{EMI}(P_{\mathbf{X}Y}; P_\theta) - \mathrm{PM}(P_{\mathbf{X}Y}; P_\theta)| &= |H(P_{Y_\theta}) - H(P_Y) + \mathbb{E}_{\mathbf{x} \sim P_{\mathbf{X}}}[\mathbb{E}_{\hat{y} \sim P_\theta(\cdot|\mathbf{x})} \log \frac{P_\theta(\hat{y}|\mathbf{x})}{P_{Y|\mathbf{X}=\mathbf{x}}(\hat{y})}]| \\
&\leq |H(P_{Y_\theta}) - H(P_Y)| + \mathbb{E}_{\mathbf{x} \sim P_{\mathbf{X}}}[D_{\mathrm{KL}}(P_\theta(\cdot|\mathbf{x}) \| P_{Y|\mathbf{X}=\mathbf{x}})] \\
&\leq 4.4\delta^{\frac{1}{8}} + \delta.
\end{aligned}
$$

$\square$

Here, we adopted Lemma D.4 to replace $|H(P_{Y_\theta}) - H(P_Y)|$ into its upper bound $4D_{\mathrm{JS}}^{\frac{1}{4}}(P_{Y_\theta} \| P_Y)$ and used Pinsker's inequality (Pinsker, 1964).

We provide a proof for the extended theorem from the Lemma D.2 by considering the optimal model parameter as below.

**Theorem D.2** (Restatement of Lemma 4.4). *Given a distribution $P_{\mathbf{X}Y}$ and an MLLM $P_\theta$, and assume $P_{\mathbf{X}Y} > c > 0$ for a constant $c$, if the $\epsilon$-representation capacity assumption holds, i.e.,*

$$\min_{\theta \in \Theta} \mathbb{E}_{\mathbf{x} \sim P_{\mathbf{X}}} D_{\mathrm{KL}}(P_{Y|\mathbf{X}=\mathbf{x}} \| P_\theta(\cdot|\mathbf{x})) \leq \epsilon, \tag{15}$$

*and let the reward function $r(\mathbf{x}, y)$ be $\log P_{Y|\mathbf{X}=\mathbf{x}}(y)$, then*

$$|EMI(P_{\mathbf{X}Y}; \theta^*) - PM(P_{\mathbf{X}Y}; \theta^*)| \leq \delta + 4.4\delta^{\frac{1}{8}},$$

*where $\theta^*$ is the optimal solution of Eq. (1) over $P_{\mathbf{X}Y}$, and $\delta = 4.4\epsilon^{\frac{1}{8}} - \log c\sqrt{2\epsilon}$.*

*Proof.* Recall the formulation of mutual information as below,

$$
\begin{aligned}
I(P_{\mathbf{X}Y}) &= H(P_Y) - H(P_{Y|\mathbf{X}}) \\
&= \mathbb{E}_{\mathbf{x}, y \sim P_{\mathbf{X}Y}}[\log P_{Y|X=x}(y)] + H(P_Y) \\
&= \mathbb{E}_{\mathbf{x}, y \sim P_{\mathbf{X}Y}}[\log P_\theta(y|\mathbf{x})] - \mathbb{E}_{\mathbf{x}, y \sim P_{\mathbf{X}Y}}[\log P_\theta(y|\mathbf{x})] + \mathbb{E}_{\mathbf{x}, y \sim P_{\mathbf{X}Y}}[\log P_{Y|X=x}(y)] + H(P_Y) \\
&= \mathbb{E}_{\mathbf{x}, y \sim P_{\mathbf{X}Y}}[\log P_\theta(y|\mathbf{x})] + \mathbb{E}_{\mathbf{x} \sim P_{\mathbf{X}}}[D_{\mathrm{KL}}(P_{Y|\mathbf{X}=\mathbf{x}} \| P_\theta(\cdot|\mathbf{x}))] + H(P_Y)
\end{aligned}
$$

So, $I(P_{\mathbf{X}Y}) - \mathbb{E}_{\mathbf{x},y \sim P_{\mathbf{X}Y}}[\log P_\theta(y|\mathbf{x})] - H(P_Y) = \mathbb{E}_{\mathbf{x} \sim P_\mathbf{X}}[D_{\mathrm{KL}}(P_{Y|\mathbf{X}=x}(y)||P_\theta(\cdot|\mathbf{x}))]$. Therefore, for the optimally learned parameter $\theta^*$, we know that $\theta^* \in \arg\min_{\theta \in \Theta} \mathbb{E}_{\mathbf{x} \sim P_\mathbf{X}}[D_{\mathrm{KL}}(P_{Y|\mathbf{X}=x}||P_\theta(\cdot|\mathbf{x}))]$, which implies below,

$$\mathbb{E}_{\mathbf{x} \sim P_\mathbf{X}}[D_{\mathrm{KL}}(P_{Y|\mathbf{X}=x}||P_{\theta^*}(\cdot|\mathbf{x}))] \leq \epsilon.$$

Meanwhile, we have the below upper bound by leveraging Lemma D.7,

$$\mathbb{E}_{\mathbf{x} \sim P_\mathbf{X}}[D_{\mathrm{KL}}(P_\theta(\cdot|\mathbf{x})||P_{Y|\mathbf{X}=\mathbf{x}})] \leq 4.4\epsilon^{\frac{1}{8}} - \log c\sqrt{2\epsilon}$$

By denoting $\delta := 4.4\epsilon^{\frac{1}{8}} - \log c\sqrt{2\epsilon}$ and plugging the Lemma D.1, we complete the proof. □

Note that the assumption $P_{\mathbf{X}Y} > c > 0$ is reasonable in practice given the following two statements. First, we can only observe the samples that $P_{\mathbf{X}Y}(\mathbf{x}, y) > 0$. Therefore, investigating on the case $\mathbf{x}, y$ such that $P_{\mathbf{X}Y} > 0$ solely does not affect the practical implication of our analysis. Second, for the space $\mathcal{X} \times \mathcal{Y}$, it is obvious that $|\mathcal{X} \times \mathcal{Y}| < +\infty$. Therefore, $\mathcal{X} \times \mathcal{Y}$ is a compact space, and $P_{\mathbf{X}Y}(\mathbf{x}, y) > 0$ over a compact space, there exists a constant $c > 0$ such that $P_{\mathbf{X}Y}(\mathbf{x}, y) > c$.

**Lemma D.3.** *Given two distributions $P_X$ and $Q_X$ defined over $\mathcal{X}$, let $f : \mathcal{X} \to [0, c]$, then we have the below,*

$$\left|\mathbb{E}_{x \sim P_X}[f(x)] - \mathbb{E}_{x \sim Q_X}[f(x)]\right| \leq c \cdot D_{\mathrm{TV}}(P_X, Q_X).$$

*where $D_{\mathrm{TV}}(P_X, Q_X) := \sum_{x \in X} |P_X(x) - Q_X(x)|$ is the total variation distance between two distributions.*

*Proof.*

$$\begin{aligned}
&\left|\mathbb{E}_{x \sim P_X}[f(x)] - \mathbb{E}_{x \sim Q_X}[f(x)]\right| \\
=&\left|\sum_{x \in \mathcal{X}} P_X(x)f(x) \sum_{x \in \mathcal{X}} Q_X(x)f(x)\right| \\
=&\left|\sum_{x \in \mathcal{X}} (P_X(x) - Q_X(x))f(x)\right| \\
=&\left|\sum_{x \in \mathcal{X}} (P_X(x) - Q_X(x))(f(x) - c) + c(\sum_{x \in \mathcal{X}} P_X(x) - Q_X(x))\right| \\
\leq&\sum_{x \in \mathcal{X}} |P_X(x) - Q_X(x)| \cdot |f(x) - c| \\
\leq&c \cdot ||P_X - Q_X||_1 \\
=&c \cdot D_{\mathrm{TV}}(P_X, Q_X)
\end{aligned}$$

□

**Lemma D.4.** *Given random variable $\mathbf{X}$, and two distributions $P_{\mathbf{X}Y} = P_{Y|\mathbf{X}}P_\mathbf{X}$ and $Q_{\mathbf{X}Y} = Q_{Y|\mathbf{X}}Q_\mathbf{X}$, we have the bounds for the difference between Entropy $H(\cdot)$ over two distributions as below:*

$$|H(P_\mathbf{X}) - H(Q_\mathbf{X})| \leq 4D_{\mathrm{JS}}^{\frac{1}{4}}(P_\mathbf{X}||Q_\mathbf{X}),$$

$$|H(P_Y) - H(Q_Y)| \leq 4D_{\mathrm{JS}}^{\frac{1}{4}}(P_Y||Q_Y),$$

$$|H(P_{Y|\mathbf{X}=\mathbf{x}}) - H(Q_{Y|\mathbf{X}=\mathbf{x}})| \leq 4D_{\mathrm{JS}}^{\frac{1}{4}}(P_{Y|\mathbf{X}=\mathbf{x}}||Q_{Y|\mathbf{X}=\mathbf{x}}).$$

*where $D_{\mathrm{JS}}(\cdot, \cdot)$ is the Jensen-Shannon divergence between two distributions.*

*Proof.* Let $M_\mathbf{X} = (P_\mathbf{X} + Q_\mathbf{X})/2$, $D_{\mathrm{JS}}(P_\mathbf{X}, Q_\mathbf{X})$ and $D_{\mathrm{TV}}(P_\mathbf{X}, Q_\mathbf{X})$ be the Jensen-Shannon divergence and total variation

distance between $P_{\mathbf{X}}$ and $Q_{\mathbf{X}}$, respectively.

$$\left|H(P_{\mathbf{X}}) - H(Q_{\mathbf{x}})\right|$$

$$=\left|\mathbb{E}_{\mathbf{x}\sim P_{\mathbf{X}}}\log P_{\mathbf{X}}(\mathbf{x}) - \mathbb{E}_{\mathbf{x}\sim Q_{\mathbf{X}}}\log Q_{\mathbf{X}}(\mathbf{x})\right|$$

$$=\left|\mathbb{E}_{\mathbf{x}\sim P_{\mathbf{X}}}\log P_{\mathbf{X}}(\mathbf{x}) - \mathbb{E}_{\mathbf{x}\sim P_{\mathbf{X}}}\log M_{\mathbf{X}}(\mathbf{x}) - \mathbb{E}_{\mathbf{x}\sim Q_{\mathbf{X}}}\log Q_{\mathbf{X}}(\mathbf{x}) + \mathbb{E}_{\mathbf{x}\sim Q_{\mathbf{X}}}\log M_{\mathbf{X}}(\mathbf{x}) + \mathbb{E}_{\mathbf{x}\sim P_{\mathbf{X}}}\log M_{\mathbf{X}}(\mathbf{x}) - \mathbb{E}_{\mathbf{x}\sim Q_{\mathbf{X}}}\log M_{\mathbf{X}}(\mathbf{x})\right|$$

$$\leq\left|\mathbb{E}_{\mathbf{x}\sim P_{\mathbf{X}}}\log \frac{P_{\mathbf{X}}(\mathbf{x})}{M_{\mathbf{X}}(\mathbf{x})} - \mathbb{E}_{\mathbf{x}\sim Q_{\mathbf{X}}}\log \frac{Q_{\mathbf{X}}(\mathbf{x})}{M_{\mathbf{X}}(\mathbf{x})}\right| + \left|\mathbb{E}_{\mathbf{x}\sim P_{\mathbf{X}}}\log M_{\mathbf{X}}(\mathbf{x}) - \mathbb{E}_{\mathbf{x}\sim Q_{\mathbf{X}}}\log M_{\mathbf{X}}(\mathbf{x})\right|$$

$$\leq\left|\mathbb{E}_{\mathbf{x}\sim P_{\mathbf{X}}}\log \frac{P_{\mathbf{X}}(\mathbf{x})}{M_{\mathbf{X}}(\mathbf{x})} + \mathbb{E}_{\mathbf{x}\sim Q_{\mathbf{X}}}\log \frac{Q_{\mathbf{X}}(\mathbf{x})}{M_{\mathbf{X}}(\mathbf{x})}\right| + \left|\mathbb{E}_{\mathbf{x}\sim P_{\mathbf{X}}}\log M_{\mathbf{X}}(\mathbf{x}) - \mathbb{E}_{\mathbf{x}\sim Q_{\mathbf{X}}}\log M_{\mathbf{X}}(\mathbf{x})\right|$$

$$\leq 2D_{\mathrm{JS}}(P_{\mathbf{X}}||Q_{\mathbf{X}}) + 2\sum_{x}|\frac{P_{\mathbf{X}}(x)}{2} - \frac{Q_{\mathbf{X}}(x)}{2}|\cdot|\log M_{\mathbf{X}}(x)|$$

$$=2D_{\mathrm{JS}}(P_{\mathbf{X}}||Q_{\mathbf{X}}) + 2\sum_{x}|\frac{P_{\mathbf{X}}(x)}{2} - \frac{Q_{\mathbf{X}}(x)}{2}|\cdot|\log|\frac{P_{\mathbf{X}}(x)}{2} + \frac{Q_{\mathbf{X}}(x)}{2}||$$

$$\leq 2D_{\mathrm{JS}}(P_{\mathbf{X}}||Q_{\mathbf{X}}) + 2\sum_{x}|\frac{P_{\mathbf{X}}(x)}{2} - \frac{Q_{\mathbf{X}}(x)}{2}|\cdot|\log|\frac{P_{\mathbf{X}}(x)}{2} - \frac{Q_{\mathbf{X}}(x)}{2}||$$

$$\leq 2D_{\mathrm{JS}}(P_{\mathbf{X}}||Q_{\mathbf{X}}) + 2\sum_{x}\sqrt{|\frac{P_{\mathbf{X}}(x)}{2} - \frac{Q_{\mathbf{X}}(x)}{2}|}$$

$$\leq 2D_{\mathrm{JS}}(P_{\mathbf{X}}||Q_{\mathbf{X}}) + \sqrt{2\sum_{x}|P_{\mathbf{X}}(x) - Q_{\mathbf{X}}(x)|}$$

$$=2D_{\mathrm{JS}}(P_{\mathbf{X}}||Q_{\mathbf{X}}) + \sqrt{2D_{\mathrm{TV}}(P_{\mathbf{X}}, Q_{\mathbf{X}})}$$

$$\leq 2D_{\mathrm{JS}}(P_{\mathbf{X}}||Q_{\mathbf{X}}) + 2D_{\mathrm{JS}}^{\frac{1}{4}}(P_{\mathbf{X}}||Q_{\mathbf{X}})$$

$$\leq 4D_{\mathrm{JS}}^{\frac{1}{4}}(P_{\mathbf{X}}||Q_{\mathbf{X}})$$

In above inequalities, we have used $\sqrt{x} + x\log x > 0$ for $x \in (0,1)$, Holder's inequality, and $D_{\mathrm{TV}}(P_{\mathbf{X}}, Q_{\mathbf{X}}) \leq \sqrt{2D_{\mathrm{JS}}(P_{\mathbf{X}}||Q_{\mathbf{X}})}$ proved in Lemma 3 of Thekumparampil et al. (2018). We can prove below with the same strategy,

$$|H(P_Y) - H(Q_Y)| \leq 4D_{\mathrm{JS}}^{\frac{1}{4}}(P_Y||Q_Y),$$

$$|H(P_{Y|\mathbf{X}=\mathbf{x}}) - H(Q_{Y|\mathbf{X}=\mathbf{x}})| \leq 4D_{\mathrm{JS}}^{\frac{1}{4}}(P_{Y|\mathbf{X}=\mathbf{x}}||Q_{Y|\mathbf{X}=\mathbf{x}}).$$

$\square$

**Corollary D.5.** *For a data distribution $P_{\mathbf{XY}} = P_{Y|\mathbf{X}}P_{\mathbf{X}}$, MLLM $P_\theta(\cdot|\mathbf{x})$, and Kullback-Leibler divergence $D_{\mathrm{KL}}$, if $\mathbb{E}_{\mathbf{x}\sim P_{\mathbf{X}}}D_{\mathrm{KL}}(P_{Y|\mathbf{X}=\mathbf{x}}||P_\theta(\cdot|\mathbf{x}))) \leq \epsilon$ for a constant $\epsilon$, then*

$$\mathbb{E}_{\mathbf{x}\sim P_{\mathbf{X}}}[H(P_\theta(\cdot|\mathbf{x})) - H(P_{Y|\mathbf{X}=\mathbf{x}})] \leq 4.4\epsilon^{\frac{1}{8}}$$

*Proof.*

$$\mathbb{E}_{\mathbf{x}\sim P_{\mathbf{X}}}[H(P_\theta(\cdot|\mathbf{x})) - H(P_{Y|\mathbf{X}=\mathbf{x}})] \leq \mathbb{E}_{\mathbf{x}\sim P_{\mathbf{X}}}[|H(P_\theta(\cdot|\mathbf{x})) - H(P_{Y|\mathbf{X}=\mathbf{x}})|]$$

$$\leq 4\mathbb{E}_{\mathbf{x}\sim P_{\mathbf{X}}}D_{\mathrm{JS}}^{\frac{1}{4}}(P_{Y|\mathbf{X}=\mathbf{x}}||P_\theta(\cdot|\mathbf{x}))$$

$$\leq 4\cdot 2^{\frac{1}{8}}\mathbb{E}_{\mathbf{x}\sim P_{\mathbf{X}}}D_{\mathrm{KL}}^{\frac{1}{8}}(P_{Y|\mathbf{X}=\mathbf{x}}||P_\theta(\cdot|\mathbf{x}))$$

$$\leq 4.4\epsilon^{\frac{1}{8}}.$$

We started from Lemma D.4 and use Pinsker's inequality (Pinsker, 1964) to leverage $D_{\mathrm{JS}}(\cdot|\cdot) \leq D_{\mathrm{TV}}(\cdot,\cdot) \leq \sqrt{2D_{\mathrm{KL}}(\cdot|\cdot)}$.

$\square$

**Lemma D.6.** *For a data distribution $P_{\mathbf{XY}} = P_{Y|\mathbf{X}}P_{\mathbf{X}}$, MLLM $P_\theta(\cdot|\mathbf{x})$, let $D_{\mathrm{KL}}$ and $D_{\mathrm{TV}}$ be Kullback-Leibler divergence and total variation distance, respectively. Denote $P_{Y_\theta} = \mathbb{E}_{\mathbf{x}\sim P_{\mathbf{X}}}[P_\theta(\cdot|\mathbf{x})]$, we have below inequality.*

$$D_{\mathrm{TV}}(P_Y, P_{Y_\theta}) \leq \sqrt{2\mathbb{E}_{\mathbf{x}\sim P_{\mathbf{X}}}D_{\mathrm{KL}}(P_\theta(\cdot|\mathbf{x})||P_{Y|\mathbf{X}=\mathbf{x}})}.$$

*Proof.*

$$D_{\mathrm{TV}}(P_Y, P_{Y_\theta}) = \sum_{y \in \mathcal{Y}} \left| \mathbb{E}_{\mathbf{x} \sim P_\mathbf{X}} P_{Y|\mathbf{X}=\mathbf{x}}(y) - \mathbb{E}_{\mathbf{x} \sim P_\mathbf{X}} P_\theta(y|\mathbf{x}) \right|$$

$$\leq \sum_{y \in \mathcal{Y}} \mathbb{E}_{\mathbf{x} \sim P_\mathbf{X}} \left| P_{Y|\mathbf{X}=\mathbf{x}}(y) - P_\theta(y|\mathbf{x}) \right|$$

$$= \mathbb{E}_{\mathbf{x} \sim P_\mathbf{X}} D_{\mathrm{TV}}(P_{Y|\mathbf{X}=\mathbf{x}}, P_\theta(\cdot|\mathbf{x}))$$

$$\leq \sqrt{2 \mathbb{E}_{\mathbf{x} \sim P_\mathbf{X}} D_{\mathrm{KL}}(P_\theta(\cdot|\mathbf{x}) \| P_{Y|\mathbf{X}=\mathbf{x}})}$$

$\square$

**Lemma D.7.** *For a data distribution $P_{\mathbf{X}Y} = P_{Y|\mathbf{X}} P_\mathbf{X}$, MLLM $P_\theta(\cdot|\mathbf{x})$, and Kullback-Leibler divergence $D_{\mathrm{KL}}$, if $\mathbb{E}_{\mathbf{x} \sim P_\mathbf{X}} D_{\mathrm{KL}}(P_{Y|\mathbf{X}=\mathbf{x}} \| P_\theta(\cdot|\mathbf{x}))) \leq \epsilon$ and $P_{\mathbf{X}Y} > c > 0$ for a constant $c$ and $\epsilon$, then*

$$\mathbb{E}_{\mathbf{x} \sim P_\mathbf{X}} D_{\mathrm{KL}}(P_\theta(\cdot|\mathbf{x}) \| P_{Y|\mathbf{X}=\mathbf{x}}) \leq 4.4 \epsilon^{\frac{1}{8}} - \log c \sqrt{2\epsilon}.$$

*Proof.* Note that

$$\mathbb{E}_{\mathbf{x} \sim P_\mathbf{X}} [H(P_\theta(\cdot|\mathbf{x})) - H(P_{Y|\mathbf{X}=\mathbf{x}})] = \mathbb{E}_{\mathbf{x} \sim P_\mathbf{X}} [\mathbb{E}_{y \sim P_\theta(\cdot|\mathbf{x})} \log P_{Y|\mathbf{X}=\mathbf{x}}(y) - \mathbb{E}_{y \sim P_{Y|\mathbf{X}=\mathbf{x}}} \log P_{Y|\mathbf{X}=\mathbf{x}}(y)]$$

$$+ \mathbb{E}_{\mathbf{x} \sim P_\mathbf{X}} D_{\mathrm{KL}}(P_\theta(\cdot|\mathbf{x}) \| P_{Y|\mathbf{X}=\mathbf{x}}).$$

Therefore,

$$\mathbb{E}_{\mathbf{x} \sim P_\mathbf{X}} D_{\mathrm{KL}}(P_\theta(\cdot|\mathbf{x})) \| P_{Y|\mathbf{X}=\mathbf{x}}) \leq \left| \mathbb{E}_{\mathbf{x} \sim P_\mathbf{X}} [\mathbb{E}_{y \sim P_\theta(\cdot|\mathbf{x})} \log P_{Y|\mathbf{X}=\mathbf{x}}(y) - \mathbb{E}_{y \sim P_{Y|\mathbf{X}=\mathbf{x}}} \log P_{Y|\mathbf{X}=\mathbf{x}}(y)] \right|$$

$$+ \mathbb{E}_{\mathbf{x} \sim P_\mathbf{X}} [H(P_\theta(\cdot|\mathbf{x})) - H(P_{Y|\mathbf{X}=\mathbf{x}})]$$

Given Lemma D.3, it is easy to check that

$$\left| \mathbb{E}_{\mathbf{x} \sim P_\mathbf{X}} [\mathbb{E}_{y \sim P_\theta(\cdot|\mathbf{x})} \log P_{Y|\mathbf{X}=\mathbf{x}}(y) - \mathbb{E}_{y \sim P_{Y|\mathbf{X}=\mathbf{x}}} \log P_{Y|\mathbf{X}=\mathbf{x}}(y)] \right|$$

$$\leq - \log c \, \mathbb{E}_{\mathbf{x} \sim P_\mathbf{X}} D_{\mathrm{TV}}(P_\theta(\cdot|\mathbf{x}), P_{Y|\mathbf{X}=\mathbf{x}})$$

$$\leq - \log c \sqrt{2\epsilon}.$$

Then, with Corollary D.5, we complete this proof. $\square$

## D.2. Proof for EMID upper bound

Now, we give the proof for the upper bound of the EMI Difference (EMID) as below.

**Theorem D.8** (General scenario). *Given $P_{\mathbf{X}Y}$ and $Q_{\mathbf{X}Y}$ distributions and an MLLM $P_\theta$, if there exist some constants $\delta_P$ and $\delta_Q$ such that*

$$D_{\mathrm{JS}}(P_{Y_\theta} \| P_Y) \leq \delta_P, \quad D_{\mathrm{JS}}(Q_{Y_\theta} \| Q_Y) \leq \delta_Q,$$

*where $P_{Y_\theta} = \mathbb{E}_{\mathbf{x} \sim P_\mathbf{X}} P_\theta(\cdot|\mathbf{x})$ and $Q_{Y_\theta} = \mathbb{E}_{\mathbf{x} \sim Q_\mathbf{X}} P_\theta(\cdot|\mathbf{x})$, then $EMID(P_{\mathbf{X}Y}, Q_{\mathbf{X}Y}; P_\theta)$ is upper bounded by*

$$\widehat{H} \left( D_{\mathrm{JS}}^{\frac{1}{2}}(P_{X_v} \| Q_{X_v}) + D_{\mathrm{JS}}^{\frac{1}{2}}(P_{X_t} \| Q_{X_t}) \right)$$

$$+ \widehat{H} \left( \bar{D}_{\mathrm{JS}}^{\frac{1}{2}}(P_{X_t|X_v} \| Q_{X_t|X_v}) + \bar{D}_{\mathrm{JS}}^{\frac{1}{2}}(P_{X_v|X_t} \| Q_{X_v|X_t}) \right)$$

$$+ 4 \mathbb{E}_{\mathbf{x} \sim P_\mathbf{X}} [D_{\mathrm{JS}}^{\frac{1}{4}}(P_{Y|\mathbf{X}=\mathbf{x}} \| Q_{Y|\mathbf{X}=\mathbf{x}})] + 8\Delta^{\frac{1}{4}},$$

*where $\Delta = \delta_P + \delta_Q$, $\widehat{H} = \max_{\mathbf{x} \in \mathcal{X}} [H(Q_{Y|\mathbf{X}=\mathbf{x}}) + H(P_\theta(\cdot|\mathbf{x}))]$ and*

$$\bar{D}_{\mathrm{JS}}(P_{X|X'} \| Q_{X|X'}) := \mathbb{E}_{\mathbf{x} \sim P_{X'}} D_{\mathrm{JS}}(P_{X|X'=\mathbf{x}} \| Q_{X|X'=\mathbf{x}}) + \mathbb{E}_{\mathbf{x} \sim Q_{X'}} D_{\mathrm{JS}}(P_{X|X'=\mathbf{x}} \| Q_{X|X'=\mathbf{x}}).$$

*Proof.* Let $P_{Y_\theta} = \mathbb{E}_{\mathbf{x} \sim P_{\mathbf{X}}} P_\theta(\cdot|\mathbf{x})$ and $Q_{Y_\theta} = \mathbb{E}_{\mathbf{x} \sim Q_{\mathbf{X}}} P_\theta(\cdot|\mathbf{x})$, EMID can be expressed with entropy and conditional entropy terms as below.

$$
\begin{aligned}
&\mathrm{EMID}(P_{\mathbf{X}Y}, Q_{\mathbf{X}Y}; P_\theta) \\
&= \mathrm{EMI}(P_{\mathbf{X}Y}; P_\theta) - \mathrm{EMI}(Q_{\mathbf{X}Y}; P_\theta) \\
&= (H(P_{Y_\theta}) - \mathbb{E}_{\mathbf{x} \sim P_{\mathbf{X}}}[H(P_\theta(\cdot|\mathbf{x}))] - H(P_Y) + H(P_{Y|\mathbf{X}})) \\
&\quad - ((H(Q_{Y_\theta}) - \mathbb{E}_{\mathbf{x} \sim Q_{\mathbf{X}}}[H(P_\theta(\cdot|\mathbf{x}))] - H(Q_Y) + H(Q_{Y|\mathbf{X}})) \\
&\leq (H(P_{Y|\mathbf{X}}) - H(Q_{Y|\mathbf{X}})) + \left(\mathbb{E}_{\mathbf{x} \sim Q_{\mathbf{X}}}[H(P_\theta(\cdot|\mathbf{x}))] - \mathbb{E}_{\mathbf{x} \sim P_{\mathbf{X}}}[H(P_\theta(\cdot|\mathbf{x}))]\right) \\
&\quad + |H(P_{Y_\theta}) - H(P_Y) + H(Q_Y) - H(Q_{Y_\theta})| \\
&\leq (H(P_{Y|\mathbf{X}}) - H(Q_{Y|\mathbf{X}})) + \left(\mathbb{E}_{\mathbf{x} \sim Q_{\mathbf{X}}}[H(P_\theta(\cdot|\mathbf{x}))] - \mathbb{E}_{\mathbf{x} \sim P_{\mathbf{X}}}[H(P_\theta(\cdot|\mathbf{x}))]\right) \\
&\quad + |H(P_{Y_\theta}) - H(P_Y)| + |H(Q_Y) - H(Q_{Y_\theta})| \\
&\leq (H(P_{Y|\mathbf{X}}) - H(Q_{Y|\mathbf{X}})) + \left(\mathbb{E}_{\mathbf{x} \sim Q_{\mathbf{X}}}[H(P_\theta(\cdot|\mathbf{x}))] - \mathbb{E}_{\mathbf{x} \sim P_{\mathbf{X}}}[H(P_\theta(\cdot|\mathbf{x}))]\right) + 4\left(D_{\mathrm{JS}}^{\frac{1}{4}}(P_{Y_\theta}, P_Y) + D_{\mathrm{JS}}^{\frac{1}{4}}(Q_{Y_\theta}, Q_Y)\right) \\
&\leq \underbrace{(H(P_{Y|\mathbf{X}}) - H(Q_{Y|\mathbf{X}}))}_{(A)} + \underbrace{\left(\mathbb{E}_{\mathbf{x} \sim Q_{\mathbf{X}}}[H(P_\theta(\cdot|\mathbf{x}))] - \mathbb{E}_{\mathbf{x} \sim P_{\mathbf{X}}}[H(P_\theta(\cdot|\mathbf{x}))]\right)}_{(B)} + 8\Delta^{\frac{1}{4}},
\end{aligned}
\tag{16}
$$

where $\Delta := \delta_P + \delta_Q$. Now, we will derive the upper-bounds for the terms (A) and (B), independently. First, by adopting Lemma D.4, we can express the term $H(P_{Y|\mathbf{X}})$ as below.

$$
\begin{aligned}
&H(P_{Y|\mathbf{X}}) \\
&= \mathbb{E}_{P_{\mathbf{X}}}[H(P_{Y|\mathbf{X}=\mathbf{x}}) - H(Q_{Y|\mathbf{X}=\mathbf{x}})] + \mathbb{E}_{P_{\mathbf{X}}}[H(Q_{Y|\mathbf{X}=\mathbf{x}})] \\
&\leq \mathbb{E}_{P_{\mathbf{X}}}[|H(P_{Y|\mathbf{X}=\mathbf{x}}) - H(Q_{Y|\mathbf{X}=\mathbf{x}})|] + \mathbb{E}_{P_{\mathbf{X}}}[H(Q_{Y|\mathbf{X}=\mathbf{x}})] \\
&\leq 4\mathbb{E}_{P_{\mathbf{X}}}[D_{\mathrm{JS}}^{\frac{1}{4}}(P_{Y|\mathbf{X}=\mathbf{x}}||Q_{Y|\mathbf{X}=\mathbf{x}})] + \mathbb{E}_{P_{\mathbf{X}}}[H(Q_{Y|\mathbf{X}=\mathbf{x}})] \\
&\leq 4\mathbb{E}_{P_{\mathbf{X}}}[D_{\mathrm{JS}}^{\frac{1}{4}}(P_{Y|\mathbf{X}=\mathbf{x}}||Q_{Y|\mathbf{X}=\mathbf{x}})] + \mathbb{E}_{P_{\mathbf{X}}}[H(Q_{Y|\mathbf{X}=\mathbf{x}})] - \mathbb{E}_{Q_{\mathbf{X}}}[H(Q_{Y|\mathbf{X}=\mathbf{x}})] + \mathbb{E}_{Q_{\mathbf{X}}}[H(Q_{Y|\mathbf{X}=\mathbf{x}})]
\end{aligned}
$$

Then, the term (A) of ineq. (16), i.e., $H(P_{Y|\mathbf{X}}) - H(Q_{Y|\mathbf{X}})$, is represented as below:

$$
H(P_{Y|\mathbf{X}}) - H(Q_{Y|\mathbf{X}}) \leq 4\mathbb{E}_{P_{\mathbf{X}}}[D_{\mathrm{JS}}^{\frac{1}{4}}(P_{Y|\mathbf{X}=\mathbf{x}}||Q_{Y|\mathbf{X}=\mathbf{x}})] + \mathbb{E}_{P_{\mathbf{X}}}[H(Q_{Y|\mathbf{X}=\mathbf{x}})] - \mathbb{E}_{Q_{\mathbf{X}}}[H(Q_{Y|\mathbf{X}=\mathbf{x}})]
$$

To replace $\mathbb{E}_{P_{\mathbf{X}}}[H(Q_{Y|\mathbf{X}=\mathbf{x}})] - \mathbb{E}_{Q_{\mathbf{X}}}[H(Q_{Y|\mathbf{X}=\mathbf{x}})]$ into a more interpretable term, and we start from the restatement of Lemma 1 of Shui et al. (2022).

**Lemma D.9** (restatement of Lemma 1 of Shui et al. (2022)). *Let $Z \in \mathcal{Z}$ be the real-valued integrable random variable, and denoting two distributions on a common space $\mathcal{Z}$ by $P$ and $Q$ such that $Q$ is absolutely continuous w.r.t. $P$. If for any function $f$ and $\lambda \in \mathbb{R}$ such that $\mathbb{E}_P[\exp(\lambda(f(z) - \mathbb{E}_P(f(z))))] < \infty$, then we have:*

$$
\begin{aligned}
\lambda(\mathbb{E}_Q f(z) - \mathbb{E}_P f(z)) &\leq D_{\mathrm{KL}}(Q||P) \\
&\quad + \log \mathbb{E}_P[\exp(\lambda(f(z) - \mathbb{E}_P(f(z))))]
\end{aligned}
$$

Now, let $\mathbf{X}$ and $Y$ denote observable variables from a joint distribution $D_{\mathbf{X}Y} \in \{P_{\mathbf{X}Y}, Q_{\mathbf{X}Y}\}$, and we denote $f(\mathbf{x}) := H(Q_{Y|\mathbf{X}=\mathbf{x}}) \geq 0$ as a loss function of our interest, e.g., conditional entropy of $y$ given $\mathbf{x}$. Then, $f$ has a finite value of $\mathbb{E}_D[\exp(\lambda(f(\mathbf{x}) - \mathbb{E}_D(f(\mathbf{x}))))]$, and is bounded within interval $[0, \hat{H}(Q_{Y|\mathbf{x}})]$ where $\hat{H}(Q_{Y|\mathbf{x}}) := \max_{\mathbf{x} \in \mathcal{X}} H(Q_{Y|\mathbf{X}=\mathbf{x}})$.

We next define a mixture distribution $M_{\mathbf{X}Y} = \frac{1}{2}(P_{\mathbf{X}Y} + Q_{\mathbf{X}Y})$ where the support of $M_{\mathbf{X}Y}$ covers that of $P_{\mathbf{X}Y}$ and $Q_{\mathbf{X}Y}$. Then, we get the inequality below by setting $P = M_{\mathbf{X}Y}$ and $Q = Q_{\mathbf{X}Y}$ for all $\lambda > 0$ according to the Lemma D.9:

$$
\begin{aligned}
&\mathbb{E}_{Q_{\mathbf{X}}}[H(Q_{Y|\mathbf{X}=\mathbf{x}})] - \mathbb{E}_{M_{\mathbf{X}}}[H(Q_{Y|\mathbf{X}=\mathbf{x}})] \\
&\leq \frac{1}{\lambda}(\log \mathbb{E}_{M_{\mathbf{X}}}[\exp(\lambda(f(\mathbf{x}) - \mathbb{E}_{M_{\mathbf{X}}}(f(\mathbf{x})))] + D_{\mathrm{KL}}(Q_{\mathbf{X}}||M_{\mathbf{X}})).
\end{aligned}
\tag{17}
$$

Also, we get similar inequality by setting $P = M_{\mathbf{X}Y}$ and $Q = P_{\mathbf{X}Y}$ for all $\lambda < 0$ according to the Lemma D.9 as below:

$$
\begin{aligned}
&\mathbb{E}_{P_{\mathbf{X}}}[H(Q_{Y|\mathbf{X}=\mathbf{x}})] - \mathbb{E}_{M_{\mathbf{X}}}[H(Q_{Y|\mathbf{X}=\mathbf{x}})] \\
&\geq \frac{1}{\lambda}(\log \mathbb{E}_{M_{\mathbf{X}}}[\exp(\lambda(f(\mathbf{x}) - \mathbb{E}_{M_{\mathbf{X}}}(f(\mathbf{x})))] + D_{\mathrm{KL}}(P_{\mathbf{X}}||M_{\mathbf{X}})).
\end{aligned}
\tag{18}
$$

Meanwhile, give that $f(\mathbf{x})$ is bounded within interval $\hat{H}(Q_{Y|\mathbf{x}})$, the $f(\mathbf{x}) - \mathbb{E}_{M_{\mathbf{x}}} f(\mathbf{x})$ becomes a sub-Gaussian (Wainwright, 2019) with the scale parameter $\sigma = \hat{H}(Q_{Y|\mathbf{x}})/2$ at most. Then, we can leverage the property of sub-Gaussian for the log moment generating function,

$$\log \mathbb{E}_{M_{\mathbf{x}}}[\exp(\lambda(f(\mathbf{x}) - \mathbb{E}_{M_{\mathbf{x}}}(f(\mathbf{x}))))] \leq \log(\exp(\frac{\lambda^2 \sigma^2}{2})) \leq \frac{\lambda^2 \hat{H}(Q_{Y|\mathbf{x}})^2}{8}. \tag{19}$$

By plugging the ineq. (19) into ineq. (17) and ineq. (18), we can derive the following new inequalities:

$$\mathbb{E}_{Q_{\mathbf{x}}}[H(Q_{Y|\mathbf{X}=\mathbf{x}})] - \mathbb{E}_{M_{\mathbf{x}}}[H(Q_{Y|\mathbf{X}=\mathbf{x}})] \leq \frac{\lambda_0 \hat{H}(Q_{Y|\mathbf{x}})^2}{8} + \frac{1}{\lambda_0} D_{\mathrm{KL}}(Q_{\mathbf{x}}||M_{\mathbf{x}}),$$

$$\mathbb{E}_{M_{\mathbf{x}}}[H(Q_{Y|\mathbf{X}=\mathbf{x}})] - \mathbb{E}_{P_{\mathbf{x}}}[H(Q_{Y|\mathbf{X}=\mathbf{x}})] \leq \frac{\lambda_0 \hat{H}(Q_{Y|\mathbf{x}})^2}{8} + \frac{1}{\lambda_0} D_{\mathrm{KL}}(P_{\mathbf{x}}||M_{\mathbf{x}}).$$

where $\lambda_0 = \lambda$ stands for $\lambda > 0$ in the ineq. (17) and $\lambda_0 = -\lambda$ for $\lambda < 0$ in the ineq. (18).

By adding both inequalities above, and setting the $\lambda_0 = \frac{2}{\hat{H}(Q_{Y|\mathbf{x}})}\sqrt{D_{\mathrm{KL}}(P_{\mathbf{X}}||M_{\mathbf{X}}) + D_{\mathrm{KL}}(Q_{\mathbf{X}}||M_{\mathbf{X}})}$ results in:

$$\mathbb{E}_{Q_{\mathbf{x}}}[H(Q_{Y|\mathbf{X}=\mathbf{x}})] - \mathbb{E}_{P_{\mathbf{x}}}[H(Q_{Y|\mathbf{X}=\mathbf{x}})] \leq \hat{H}(Q_{Y|\mathbf{x}})\sqrt{2D_{\mathrm{JS}}(P_{\mathbf{X}}||Q_{\mathbf{X}})}. \tag{20}$$

Next, a decomposition of KL divergence and the definition of JS divergence leads to the following inequality,

$$2D_{\mathrm{JS}}(P_{X_v X_t}||Q_{X_v X_t})$$
$$= D_{\mathrm{KL}}(P_{X_v X_t}||M_{X_v X_t}) + D_{\mathrm{KL}}(Q_{X_v X_t}||M_{X_v X_t})$$
$$= \frac{1}{2}\left(D_{\mathrm{KL}}(P_{X_v}||M_{X_v}) + \mathbb{E}_{P_{X_v}} D_{\mathrm{KL}}(P_{X_t|X_v=x_v}||M_{X_t|X_v=x_v})\right)$$
$$+ \frac{1}{2}\left(D_{\mathrm{KL}}(Q_{X_v}||M_{X_v}) + \mathbb{E}_{Q_{X_v}} D_{\mathrm{KL}}(Q_{X_t|X_v=x_v}||M_{X_t|X_v=x_v})\right)$$
$$+ \frac{1}{2}\left(D_{\mathrm{KL}}(P_{X_t}||M_{X_t}) + \mathbb{E}_{P_{X_t}} D_{\mathrm{KL}}(P_{X_v|X_t=x_t}||M_{X_v|X_t=x_t})\right)$$
$$+ \frac{1}{2}\left(D_{\mathrm{KL}}(Q_{X_t}||P_{X_t}) + \mathbb{E}_{Q_{X_t}} D_{\mathrm{KL}}(Q_{X_v|X_t=x_t}||M_{X_v|X_t=x_t})\right)$$
$$= D_{\mathrm{JS}}(P_{X_v}||Q_{X_v}) + D_{\mathrm{JS}}(P_{X_t}||Q_{X_t})$$
$$+ \frac{1}{2}\left(\mathbb{E}_{P_{X_v}} D_{\mathrm{KL}}(P_{X_t|X_v=x_v}||M_{X_t|X_v=x_v}) + \mathbb{E}_{Q_{X_v}} D_{\mathrm{KL}}(Q_{X_t|X_v=x_v}||M_{X_t|X_v=x_v})\right)$$
$$+ \frac{1}{2}\left(\mathbb{E}_{P_{X_t}} D_{\mathrm{KL}}(P_{X_v|X_t=x_t}||M_{X_v|X_t=x_t}) + \mathbb{E}_{Q_{X_t}} D_{\mathrm{KL}}(Q_{X_v|X_t=x_t}||M_{X_v|X_t=x_t})\right)$$
$$\leq D_{\mathrm{JS}}(P_{X_v}||Q_{X_v}) + D_{\mathrm{JS}}(P_{X_t}||Q_{X_t}) + \bar{D}_{\mathrm{JS}}(P_{X_t|X_v}||Q_{X_t|X_v}) + \bar{D}_{\mathrm{JS}}(P_{X_v|X_t}||Q_{X_v|X_t})$$

where $\bar{D}_{\mathrm{JS}}(P_{Y|X}||Q_{Y|X}) := \mathbb{E}_{x \sim P_X} D_{\mathrm{JS}}(P_{Y|X=x}||Q_{Y|X=x}) + \mathbb{E}_{x \sim Q_X} D_{\mathrm{JS}}(P_{Y|X=x}||Q_{Y|X=x})$

Based on the above decomposition, we can modify the bound as below,

$$\mathbb{E}_{Q_{\mathbf{x}}}[H(Q_{Y|\mathbf{X}=\mathbf{x}})] - \mathbb{E}_{P_{\mathbf{x}}}[H(Q_{Y|\mathbf{X}=\mathbf{x}})]$$
$$\leq \hat{H}(Q_{Y|\mathbf{x}})\sqrt{2D_{\mathrm{JS}}(P_{\mathbf{X}}||Q_{\mathbf{X}})}$$
$$\leq \hat{H}(Q_{Y|\mathbf{x}})\sqrt{D_{\mathrm{JS}}(P_{X_v}||Q_{X_v}) + D_{\mathrm{JS}}(P_{X_t}||Q_{X_t}) + \bar{D}_{\mathrm{JS}}(P_{X_t|X_v}||Q_{X_t|X_v}) + \bar{D}_{\mathrm{JS}}(P_{X_v|X_t}||Q_{X_v|X_t})}.$$

Therefore, we get an upper-bound for the (A) term in ineq. (16) as below,

$$H(P_{Y|\mathbf{x}}) - H(Q_{Y|\mathbf{x}})$$
$$\leq H(Q_Y)\sqrt{D_{\mathrm{JS}}(P_{X_v}||Q_{X_v}) + D_{\mathrm{JS}}(P_{X_t}||Q_{X_t}) + \bar{D}_{\mathrm{JS}}(P_{X_t|X_v}||Q_{X_t|X_v}) + \bar{D}_{\mathrm{JS}}(P_{X_v|X_t}||Q_{X_v|X_t})}$$
$$+ 4\mathbb{E}_{P_\mathbf{x}}[D_{\mathrm{JS}}^{\frac{1}{4}}(P_{Y|\mathbf{X}=\mathbf{x}}||Q_{Y|\mathbf{X}=\mathbf{x}})]$$
$$\leq \hat{H}(Q_{Y|\mathbf{x}})\big(D_{\mathrm{JS}}^{\frac{1}{2}}(P_{X_v}||Q_{X_v}) + D_{\mathrm{JS}}^{\frac{1}{2}}(P_{X_t}||Q_{X_t})\big)$$
$$+ \hat{H}(Q_{Y|\mathbf{x}})\big(\bar{D}_{\mathrm{JS}}^{\frac{1}{2}}(P_{X_t|X_v}||Q_{X_t|X_v}) + \bar{D}_{\mathrm{JS}}^{\frac{1}{2}}(P_{X_v|X_t}||Q_{X_v|X_t})\big)$$
$$+ 4\mathbb{E}_{P_\mathbf{x}}[D_{\mathrm{JS}}^{\frac{1}{4}}(P_{Y|\mathbf{X}=\mathbf{x}}||Q_{Y|\mathbf{X}=\mathbf{x}})].$$

Then, deriving a bound for the remaining (B) term in ineq. (16), i.e., $\mathbb{E}_{\mathbf{x}\sim Q_\mathbf{x}}[H(P_\theta(\cdot|\mathbf{x}))] - \mathbb{E}_{\mathbf{x}\sim P_\mathbf{x}}[H(P_\theta(\cdot|\mathbf{x}))]$, is very similar to the procedure of deriving the upper-bound for the term (A) by switching $Q_{Y|\mathbf{X}}$ to $P_\theta(\cdot|\mathbf{X})$ and set the $f$ for Lemma D.9 as $f(\mathbf{x}) := H(P_\theta(\cdot|\mathbf{x}))$, thereby having the interval $[0, \hat{H}(P_\theta)]$ where $\hat{H}(P_\theta) : \max_{\mathbf{x}\in\mathcal{X}} H(P_\theta(\cdot|\mathbf{x}))$. This induces an upper-bound as below,

$$\mathbb{E}_{\mathbf{x}\sim Q_\mathbf{x}}[H(P_\theta(\cdot|\mathbf{x}))] - \mathbb{E}_{\mathbf{x}\sim P_\mathbf{x}}[H(P_\theta(\cdot|\mathbf{x}))]$$
$$\leq \hat{H}(P_\theta)\sqrt{2D_{\mathrm{JS}}(P_\mathbf{X}||Q_\mathbf{X})} \tag{21}$$
$$\leq \hat{H}(P_\theta)\big(D_{\mathrm{JS}}^{\frac{1}{2}}(P_{X_v}||Q_{X_v}) + D_{\mathrm{JS}}^{\frac{1}{2}}(P_{X_t}||Q_{X_t}) + \bar{D}_{\mathrm{JS}}^{\frac{1}{2}}(P_{X_t|X_v}||Q_{X_t|X_v}) + \bar{D}_{\mathrm{JS}}^{\frac{1}{2}}(P_{X_v|X_t}||Q_{X_v|X_t})\big).$$

Finally, we complete the proof by adding the ineq. (D.2) and ineq. (21) to induces the upper bound of $\mathrm{EMID}(P_{\mathbf{X}Y}, Q_{\mathbf{X}Y}; P_\theta)$,

$$\mathrm{EMID}(P_{\mathbf{X}Y}, Q_{\mathbf{X}Y}; P_\theta)$$
$$\leq (H(P_{Y|\mathbf{X}}) - H(Q_{Y|\mathbf{X}})) + \big(\mathbb{E}_{\mathbf{x}\sim Q_\mathbf{x}}[H(P_\theta(\cdot|\mathbf{x}))] - \mathbb{E}_{\mathbf{x}\sim P_\mathbf{x}}[H(P_\theta(\cdot|\mathbf{x}))]\big) + 8\Delta^{\frac{1}{4}}$$
$$\leq \hat{H}\big(D_{\mathrm{JS}}^{\frac{1}{2}}(P_{X_v}||Q_{X_v}) + D_{\mathrm{JS}}^{\frac{1}{2}}(P_{X_t}||Q_{X_t})\big)$$
$$+ \hat{H}\big(\bar{D}_{\mathrm{JS}}^{\frac{1}{2}}(P_{X_t|X_v}||Q_{X_t|X_v}) + \bar{D}_{\mathrm{JS}}^{\frac{1}{2}}(P_{X_v|X_t}||Q_{X_v|X_t})\big)$$
$$+ 4\mathbb{E}_{P_\mathbf{x}}[D_{\mathrm{JS}}^{\frac{1}{4}}(P_{Y|\mathbf{X}=\mathbf{x}}||Q_{Y|\mathbf{X}=\mathbf{x}})] + 8\Delta^{\frac{1}{4}}$$

where $\widehat{H} = \max_{\mathbf{x}\in\mathcal{X}}[H(Q_{Y|\mathbf{X}=\mathbf{x}}) + H(P_\theta(\cdot|\mathbf{x}))]$ and $\Delta = \delta_P + \delta_Q$. $\qquad\square$

Then, we introduce an assumption over the consistency between conditional distributions as below.

**Assumption D.10** (Consistency of conditional distributions). *For the distributions $P_{\mathbf{X}Y}$ and $Q_{\mathbf{X}Y}$ over $\mathcal{X} \times \mathcal{Y}$, conditional distributions of $X_t$ given $X_v$, $X_v$ given $X_t$, and $Y$ given $\mathbf{X} = (X_v, X_t)$ are consistent between $P_{\mathbf{X}Y}$ and $Q_{\mathbf{X}Y}$. That is,*

- $P_{X_t|X_v} = Q_{X_t|X_v}$ *and* $P_{X_v|X_t} = Q_{X_v|X_t}$,

- $P_{Y|\mathbf{X}} = Q_{Y|\mathbf{X}}$.

Finally, we present the simplified version of EMID upper bound by leveraging the Assumption D.10.

**Theorem D.11** (Simplified scenario). *Given an MLLM $P_\theta$ and distributions $P_{\mathbf{X}Y}$, $Q_{\mathbf{X}Y}$ which have consistent conditional distributions over variables $X_v|X_t$, $X_t|X_v$, and $Y|\mathbf{X}$, if there exist some constants $\delta_P$ and $\delta_Q$ such that*

$$D_{\mathrm{JS}}(P_{Y_\theta}||P_Y) \leq \delta_P, \quad D_{\mathrm{JS}}(Q_{Y_\theta}||Q_Y) \leq \delta_Q,$$

*where $P_{Y_\theta} = \mathbb{E}_{\mathbf{x}\sim P_\mathbf{X}} P_\theta(\cdot|\mathbf{x})$ and $Q_{Y_\theta} = \mathbb{E}_{\mathbf{x}\sim Q_\mathbf{X}} P_\theta(\cdot|\mathbf{x})$, then $\mathrm{EMID}(P_{\mathbf{X}Y}, Q_{\mathbf{X}Y}; P_\theta)$ is upper bounded by*

$$\widehat{H}\big(D_{\mathrm{JS}}^{\frac{1}{2}}(P_{X_v}||Q_{X_v}) + D_{\mathrm{JS}}^{\frac{1}{2}}(P_{X_t}||Q_{X_t})\big) + 8\Delta^{\frac{1}{4}}, \tag{22}$$

*where $\widehat{H} = \max_{\mathbf{x}\in\mathcal{X}}[H(Q_{Y|\mathbf{X}=\mathbf{x}}) + H(P_\theta(\cdot|\mathbf{x}))]$ and $\Delta = \delta_P + \delta_Q$.*

*Proof.* Given Theorem D.8, Assumption D.10 zeros out the terms $\left( \bar{D}_{\mathrm{JS}}^{\frac{1}{2}}(P_{X_t|X_v} \| Q_{X_t|X_v}) + \bar{D}_{\mathrm{JS}}^{\frac{1}{2}}(P_{X_v|X_t} \| Q_{X_v|X_t}) \right)$ and $\mathbb{E}_{\mathbf{x} \sim P_{\mathbf{X}}}[D_{\mathrm{JS}}^{\frac{1}{4}}(P_{Y|\mathbf{X}=\mathbf{x}} \| Q_{Y|\mathbf{X}=\mathbf{x}})]$ which induces Eq. (22), accordingly. $\qquad\square$

**Corollary D.12.** *Given an MLLM $P_\theta$ and distributions $P_{\mathbf{X}Y}$, $Q_{\mathbf{X}Y}$ which have consistent conditional distributions over variables $X_v|X_t$, $X_t|X_v$, and $Y|\mathbf{X}$, EMID$(P_{\mathbf{X}Y}, Q_{\mathbf{X}Y}; P_\theta)$ is upper bounded by*

$$
\begin{aligned}
&\widehat{H}\left( D_{\mathrm{JS}}^{\frac{1}{2}}(P_{X_v} \| Q_{X_v}) + D_{\mathrm{JS}}^{\frac{1}{2}}(P_{X_t} \| Q_{X_t}) \right) \\
&+ 8 \left( \mathbb{E}_{\mathbf{x} \sim P_{\mathbf{X}}} D_{\mathrm{TV}}(P_{Y|\mathbf{X}=\mathbf{x}}, P_\theta(\cdot|\mathbf{x})) + \mathbb{E}_{\mathbf{x} \sim Q_{\mathbf{X}}} D_{\mathrm{TV}}(Q_{Y|\mathbf{X}=\mathbf{x}}, P_\theta(\cdot|\mathbf{x})) \right)^{\frac{1}{4}},
\end{aligned}
\tag{23}
$$

*where $D_{\mathrm{TV}}(\cdot, \cdot)$ is the total variation distance, and $\widehat{H} = \max_{\mathbf{x} \in \mathcal{X}}[H(Q_{Y|\mathbf{X}=\mathbf{x}}) + H(P_\theta(\cdot|\mathbf{x}))]$.*

*Proof.* Let $P_{Y_\theta} = \mathbb{E}_{\mathbf{x} \sim P_{\mathbf{X}}} P_\theta(\cdot|\mathbf{x})$ and $Q_{Y_\theta} = \mathbb{E}_{\mathbf{x} \sim Q_{\mathbf{X}}} P_\theta(\cdot|\mathbf{x})$, then $D_{\mathrm{JS}}(\cdot\|\cdot) \le D_{\mathrm{TV}}(\cdot, \cdot)$ allow us to induce below,

$$
\begin{aligned}
D_{\mathrm{JS}}(P_{Y_\theta} \| P_Y) &= \mathbb{E}_{\mathbf{x} \sim P_{\mathbf{X}}}[D_{\mathrm{JS}}(P_\theta(\cdot|\mathbf{x}) \| P_{Y|\mathbf{X}=\mathbf{x}})] \le \mathbb{E}_{\mathbf{x} \sim P_{\mathbf{X}}}[D_{\mathrm{TV}}(P_{Y|\mathbf{X}=\mathbf{x}}, P_\theta(\cdot|\mathbf{x}))], \\
D_{\mathrm{JS}}(Q_{Y_\theta} \| Q_Y) &= \mathbb{E}_{\mathbf{x} \sim Q_{\mathbf{X}}}[D_{\mathrm{JS}}(P_\theta(\cdot|\mathbf{x}) \| Q_{Y|\mathbf{X}=\mathbf{x}})] \le \mathbb{E}_{\mathbf{x} \sim Q_{\mathbf{X}}}[D_{\mathrm{TV}}(Q_{Y|\mathbf{X}=\mathbf{x}}, P_\theta(\cdot|\mathbf{x}))].
\end{aligned}
\tag{24}
$$

Noting that $a^{\frac{1}{4}} + b^{\frac{1}{4}} \le 2(a+b)^{\frac{1}{4}}$ for $a, b \ge 0$, plugging the above inequality into the Theorem D.11 complete the proof. $\qquad\square$

Although this alternative upper bound is looser than Theorem D.11, Corollary D.12 is more interpretable in the sense that it directly represents the model-specific discrepancy terms via distance between model output distribution and true conditional distributions, rather than expresses it through marginal distribution terms in $D_{\mathrm{JS}}(P_{Y_\theta} \| P_Y)$ and $D_{\mathrm{JS}}(Q_{Y_\theta} \| Q_Y)$. Therefore, as our model becomes more accurate at modeling the ground truth conditional distribution of $Y$ given $\mathbf{X}$, the EMID mainly depends on the divergence between the marginal distributions of visual and text inputs.

### D.3. Discussion

We made the $\epsilon$-representation capacity assumption to derive Lemma 4.3 and Theorem 4.4 that claim the theoretical justification for EMI by showing its connection to a classic preference model. The assumption captures a minimum achievable discrepancy between the truth distribution $P_{Y|\mathbf{X}}$ and the model's distribution $P_\theta(\cdot|\mathbf{x})$. As the models become more expressive–e.g., by increasing model size (Kaplan et al., 2020) and leveraging advanced positional encoding (Luo et al., 2022)–MLLM approaches the universal approximator of sequence-to-sequence mapping (Luo et al., 2022; Furuya et al., 2024), as a result, the minimum expected discrepancy tends to decrease, leading to a smaller $\epsilon$.

Meanwhile, EMI and EMID trigger flexible potential use cases given their generality. For example, they can be used as metrics to evaluate a pure LLM as well as a multimodal LLM, depending on the types of targeted problems. Moreover, although we confined our analysis on $I(P_{\mathbf{X}Y})$, the chain rule of the mutual information (Cover, 1999) allows us to conduct partial modality EMI analysis through $I(P_{\mathbf{X}Y}) = 0.5I(P_{X_v Y}) + 0.5I(P_{X_t Y}) + 0.5I(P_{X_t Y|X_v}) + 0.5I(P_{X_v Y|X_t})$, where we can decompose the aggregated EMI into the modality-specific EMI terms and modality interaction (conditional) EMI terms.

