# OpenReview forum: "Understanding Multimodal LLMs Under Distribution Shifts: An Information-Theoretic Approach"
_ICML.cc/2025/Conference — ICML 2025 poster_

### Official Review · Reviewer_vm1D · 2025-03-11

**Overall Recommendation:** 3

**Summary:**

This paper introduces an information-theoretic framework to analyze and understand the performance of Multimodal Large Language Models (MLLMs) under distribution shifts, which occur when the evaluation data differs from the instruction tuning distribution. The authors propose the concept of Effective Mutual Information (EMI), a metric that quantifies the relevance between input queries and model responses. EMI provides a more robust and theoretically grounded alternative to existing empirical metrics like the win rate. The paper derives upper bounds for the EMI difference between in-distribution (ID) and out-of-distribution (OOD) data, linking it to visual and textual distributional discrepancies. Through extensive experiments across 61 real-world distribution shift scenarios, the authors validate their framework, demonstrating strong correlations between EMI and model performance. The results suggest that EMI can be used for more reliable and cost-effective evaluation of MLLMs, particularly in situations where traditional evaluation methods may be computationally expensive or less transparent.

**Claims And Evidence:**

Based on the results in Figure 1, the authors claim that as the severity of the shift increases, the performance degradation becomes more significant. First, part of the result is covered by the legend in text shift LlaVA v1.5, which requires some alteration. Second, the result in text shift does not clearly verify the statement. According to the context, the x-axis is sorted by severity of shifts, but the win rate in text shift isn’t decreasing monotonously. Also, LlaVA NeXT 7B performs better in Korean than in Arbaic, yet LlaVA NeXT 13B shows opposite performance respectively.

**Essential References Not Discussed:**

No.

**Experimental Designs Or Analyses:**

The results in terms of Spearman correlation analysis and Kendall’s tau analysis needs further explanation, it is beneficial to specify the meaning of the value for better understanding.
In Figure 3, all four model results are shown in one graph, making it difficult to understand the authors’ implications. Also, from the figure, it seems that EMID is a lot smaller than EMID upper bound, for example, in synthetic shift graph, EMID ranges from -0.02 to 0.10, but the upper bound ranges from 1.1 to 1.8. If not misunderstood, this shows that the upper bound is very loose and has little constraint on EMID.

**Methods And Evaluation Criteria:**

Yes.

**Other Comments Or Suggestions:**

No.

**Other Strengths And Weaknesses:**

Strengths
- Understanding MLLMs under distribution shifts seems a critical research problem.
- The paper is well constructed and easy to read.

Weaknesses
- In Figure 1, the authors claim that as shift severity increases, performance degradation worsens. However, part of the result in the text shift is obscured by the legend for LLaVA v1.5, requiring adjustment. Additionally, the text shift results do not fully support this claim, as the win rate does not decrease monotonically with shift severity. For instance, LLaVA NeXT 7B performs better in Korean than Arabic, while LLaVA NeXT 13B shows the opposite trend.
- The results from the Spearman correlation analysis and Kendall’s tau analysis require further clarification. It would be helpful to specify the meaning of the correlation values to aid in better understanding their implications.
- In Figure 3, combining all four models in one graph makes it hard to grasp the authors’ implications. Additionally, the EMID appears much smaller than its upper bound—e.g., in the synthetic shift graph, EMID ranges from -0.02 to 0.10, while the upper bound spans from 1.1 to 1.8. If not misunderstood, this suggests the upper bound is very loose and offers little constraint on EMID.

**Questions For Authors:**

Please refer to the weakness part.

**Relation To Broader Scientific Literature:**

This paper introduces Effective Mutual Information (EMI) to evaluate MLLMs under distribution shifts, extending information-theoretic frameworks to multimodal models. It provides a formal method to quantify performance degradation, addressing gaps in out-of-distribution generalization and model robustness in real-world applications.

**Theoretical Claims:**

Yes.

---

> ### Author Rebuttal · Authors · 2025-03-31
>
> > _A1. Legend issue in Fig. 1. and non-monotonic performance trend in text shifts_.
>
> Thank you for pointing out the visualization issue—we will revise the legend in Fig. 1 to improve clarity and avoid confusion!
>
> * Regarding the non-monotonic trend in win rate under text shifts, this behavior arises in part from the inherent stochasticity of win rate computation based on GPT-4 API evaluations, making it fundamentally difficult to observe strictly monotonic trends in practice [1]. Additionally, the x-axis in Fig. 1 is sorted by embedding space distances—computed using CLIP ViT for visual shifts and XLM-RoBERTa for text shifts—which may not always reflect the true degree of distributional shift.
> * That said, we still observe a meaningful overall relationship between embedding distance and performance degradation, both in the 27 natural shifts presented in Fig. 1 and in the 34 synthetic shifts shown in Fig. 6 of the Appendix. By taking inspiration from this empirical analysis, we derive a much more rigorous framework, i.e., EMID upper bound (Theorem 4.5), to quantify the performance gap that consistently shows statistical significance across diverse settings.
>
> > _A2. Clarification for the meaning of Spearman correlation and Kendall’s tau_.
>
> * Spearman's $\rho$ and Kendall’s $\tau$ are both representative measures for monotonic relationships between two variables, where the former is preferred for detecting weak correlation and the latter is preferred to capture strong correlation in small sample sizes and is more robust to outliers with large sample sizes. Both of them are standard approaches to measure the correlation between LLM Judge score and other metrics [2], and are good for measuring the relation of two variables, even if they have different data types, e.g., discrete (win rate) versus continuous (EMI).
> * Both correlation coefficients range from -1.0 (negative correlation) to 1.0 (positive correlation), where 0.0 indicates there is no monotonic relationship between two variables. For Spearman's $\rho$, a 0.2-0.4 range of values denotes weak correlation, 0.4-0.6 denotes moderate correlation, and 0.6-0.8 and 0.8-1.0 denote strong and very strong correlations, respectively, and Kendall's $\tau$ can be similarly interpreted by multiplying 1.5, i.e., $\rho=1.5 \tau$, to compensate its relatively smaller scale in practice.
> * Our analysis in the paper (Table 2) indicates that the EMI consistently shows moderate or strong correlation with the LLM-judge evaluation metric, win rate, across different types of shifts and model architectures.
>
> We will add this description in our future version of the manuscript. Thank you for the suggestion.
>
> > _A3. Intricate visualization of Figure 3 and the tightness of EMID upper bound_.
> * On the left two panels in Figure 3, we presented the overall relationship between EMID and its upper bound, whereas **we distinguished four different models on the right two panels to show the model-dependent difference over the relationship**. We did so to show that Theorem 4.5 actually differentiates the model itself through the output entropy of each model $H(P_{\theta}(\cdot|x))$, i.e., LLaVA NeXT shows higher sensitivity to the shifts compared with LLaVA v1.5 implied by the larger slope.
> * In this work, we do not claim the tightness of the derived upper bound. Moreover, verifying the tightness of the proposed bound can be affected by the choice of estimators for the MI and JSD terms during empirical validation. However, we would like to emphasize that **the bound shows consistent correlation with statistical significance across 61 cases of distribution shift over four different models, which means that our analytic bound of EMID effectively stands for performance degradation of MLLM.** We appreciate your valuable concern, and we will explore devising a much tighter bound in future work.
>
> > Reference
> 1. Judging LLM-as-a-Judge with MT-Bench and Chatbot Arena, Zheng et al. 2023
> 2. PROMETHEUS: INDUCING FINE-GRAINED EVALUATION CAPABILITY IN LANGUAGE MODELS, Kim et al. 2024

---

> > ### Comment · Reviewer_vm1D · 2025-04-09
> >
> > Thank you for your detailed response. I have also read the comments from other reviewers. Most of my concerns have been adequately addressed.  As a result, I would like to keep my score as '3', leading to acceptance.

---

> > > ### Author Response · Authors · 2025-04-09
> > >
> > > Dear Reviewer vm1D,
> > >
> > >
> > > The authors would like to express their sincere gratitude to Reviewer vm1D for taking valuable time to read and provide helpful feedback that contributes to improving the quality of this work! We are happy to hear that most of the concerns were adequately addressed, and thanks for acknowledging our contribution.
> > >
> > > Best regards,
> > > The authors

---

### Official Review · Reviewer_cWre · 2025-03-12

**Overall Recommendation:** 3

**Summary:**

This paper introduces Effective Mutual Information (EMI), a novel metric for quantifying the relevance between input queries and model responses in MLLMs. Unlike standard Mutual Information, EMI removes domain-dependent components, making it a more generalizable measure especially in OOD settings. The authors establish theoretical connections between EMI and win rate, providing an intuitive explanation of EMI’s effectiveness. Additionally, they derive an upper bound for the performance gap in OOD scenarios using EMID. Extensive experiments validate EMI’s utility, confirming its theoretical predictions and demonstrating its effectiveness across diverse distribution shift scenarios.

**Claims And Evidence:**

yes

**Essential References Not Discussed:**

n/a

**Experimental Designs Or Analyses:**

yes

**Methods And Evaluation Criteria:**

yes

**Other Comments Or Suggestions:**

n/a

**Other Strengths And Weaknesses:**

Strengths:

1. This paper provides a theoretical framework to analyze and understand the impact of distribution shifts on MLLM performance via EMI.
2. The intuition of this new metric EMI is explained via analogy with excess risk, effective robustness and win rate.
3. The empirical findings strongly support most of the theoretical conclusions.

Weaknesses:

1. I appreciate the theoretical part. However,  the current theories do not address how modality fusion and modality interaction affect generalization, despite these techniques being commonly employed in  (MLLMs).
2. In Theorem 4.4, the paper assume that the $\epsilon$-representation capacity  holds, where itself requires to be quantified, e.g. it it closed related to the model size, the number of training samples  and the input dimension.  Given that the assumption fundementally underpins the theoretical contributions presented, I suggest expanding the discourse to elucidate the quantitative interdependencies between   $\epsilon$-representation capacity and these factors.
3. The paper does not demonstrate how to leverage EMI or the EMID upper bound to guide model optimization (e.g., designing robust training objectives or adaptation strategies). It only uses EMI as a post-hoc evaluation tool.

**Questions For Authors:**

1. The paper states, “In Eq. (4), we show that the autoregressive objective for instruction tuning (Eq. (1)) effectively estimates the lower bound of MI when the model’s representation capacity is sufficiently high.” However, while the term 𝛿 can be omitted under this assumption, I suppose that 𝐻(𝑃𝑌) could be significantly large, making Eq. (4) an inaccurate estimate of the MI lower bound.

2. The computation of EMI relies on pre-trained encoders (e.g., CLIP and XLM-R) for feature extraction, but the paper does not discuss the sensitivity of these encoders to domain shifts. For instance, CLIP may underperform on medical images, leading to distorted EMI estimates.

**Relation To Broader Scientific Literature:**

above average

**Theoretical Claims:**

yes

---

> ### Author Rebuttal · Authors · 2025-03-31
>
> > _A1. Effect of modality fusion/interaction on generalization_.
>
> * MLLMs commonly undergo a modality alignment phase during training, which may affect generalization, and it is known that modality fusion can reduce the sample complexity to improve generalization [1]!
> * As noted in `line 250-254` in our paper, **$I(P_{\mathbf{X}Y})$ can be factorized into $0.5 I(P_{X_{v}Y}) + 0.5 I(P_{X_{t}Y}) + 0.5 I(P_{X_{t}Y|X_v}) + 0.5 I(P_{X_{v}Y|X_t})$ where the conditional MI terms $I(P_{X_{t}Y|X_v})$ and $I(P_{X_{v}Y|X_t})$ encapsulate modality interaction.**
> * Based on this factorization, we can define per-modality EMI based on $I(P_{X_{v}Y})$ and $I(P_{X_{t}Y})$, and then derive a new upper bound that is constructed with $I(P_{X_{t}Y|X_v})$ and $I(P_{X_{v}Y|X_t})$ terms to capture the effect of modality interaction. We leave the explicit derivation for future work.
>
> > _A2. Interdependencies between $\epsilon$-representation capacity and the model size, number of training samples, and input dimension_.
>
> The $\epsilon$-representation capacity assumption captures the minimum achievable discrepancy between truth distribution $P_{Y|X}$ and the model's distribution $P_{\theta}(\cdot|X)$. Due to the expectation and $\min$ operator, it does not depend on the training sample size but is mainly influenced by model capability.
>
> Specifically, as the models become more expressive—e.g., by increasing model size [5] and leveraging advanced positional encoding [6]—MLLM approaches the universal approximator of sequence-to-sequence mapping [6,7], as a result, the minimum expected discrepancy tends to decrease, leading to a smaller $\epsilon$.
>
> We will elucidate this in the next version, thanks!
>
> > _A3. How to leverage EMID upper bound to guide model optimization?_
>
> While our primary focus is on presenting the theoretical framework to quantify the performance gap of MLLMs, we also showcase an application of the EMID upper bound in this rebuttal, a **regularization term for visual instruction tuning**.
>
> Due to the space limit, we have included the setups and results in the `rebuttal to reviewer o1Fb, response A3`. **Please refer to that thread!** As shown in the tables, our instantiation of Theorem 4.5 can indeed be used to optimize model to improve robustness under shifts.
>
> > _A4. Eq. (4) can be an inaccurate estimate of MI lower bound due to potentially large $H(P_Y)$_.
> * In the Eq. (4): $I(P_{XY})\geq\mathbb{E}[\log P_{\theta}(y|x)]+H(P_Y)$, maximizing $\mathbb{E}[\log P_{\theta}(y|x)]$ through instruction tuning can be interpreted to learn a parameter $\theta$ that maximizes $I(P_{XY})-H(P_Y)$ rather than solely $I(P_{XY})$.
> * **We do not claim that the log-likelihood term is a tight lower bound of MI but rather suggest that maximizing $\mathbb{E}[\log P_{\theta}(y|x)]$ can implicitly maximize the MI between input and model's response**. We will revise the paper to make this clear.
> * To validate this, we reproduce the visual instruction tuning of LLaVA-v1.5-7B on a 10% subset of data, and show how the empirical estimate [2] of MI evolves during training.
>     |Step|$\hat{I}$|
>     |-|-|
>     |1|0.166|
>     |5|0.172|
>     |20|0.182|
>     |100|0.187|
>     |200|0.194|
>     |500|0.197|
> * As shown above, visual instruction tuning can effectively maximize MI between input and model response.
>
> > _A5. Encoder sensitivity analysis to domain shifts for EMI estimation._
>
> * In **Table 4 and 5 of Appendix**, we already discussed two alternative choices of the encoders, e.g., [3], and showed that _our theoretical claims hold in practice consistently across varying encoders with statistical significance_.
> * We further conduct encoder sensitivity analysis under domain shifts by replicating experiments on medical domains with a CLIP-ViT-B32 and XLM-RoBERTa encoders.
> * Specifically, we use 200 samples of LLaVA-Med [4], get three splits of them based on embedding distance with COCO images, and translate English queries into six different languages used in the paper by using GPT-4o to induce 28 subsets of shifts to conduct correlation analysis for EMID and its upper bound.
>   |Model|Pearson $r$|$p$-val|
>   |-|-|-|
>   |LLaVA-v1.5.7B |0.93|0.00|
> * We see the correlation between EMID and its upper bound estimates is very strong, even though the medical image and text are relatively minority instances compared with general object and text, implying that our theorem robustly holds even on the special domains that encoders may not excel at.
>
> > Reference
> 1. A Theory of Multimodal Learning, Lu 2023
> 2. A Contrastive Log-ratio Upper Bound of Mutual Information, Cheng et al. 2020
> 3. Universal Embeddings with Multimodal Large Language Models, Jiang et al. 2024
> 4. LLaVA-Med: Training a Large Language-and-Vision Assistant for Biomedicine in One Day, Li et al. 2023
> 5. Scaling Laws for Neural Language Models, Kaplan et al. 2020
> 6. Your Transformer May Not be as Powerful as You Expect, Luo et al. 2022
> 7. Transformers are Universal In-context Learners, Furuya et al. 2024

---

### Official Review · Reviewer_sTSP · 2025-03-13

**Overall Recommendation:** 2

**Summary:**

I struggled to understand the paper so take this with a grain of salt:

The authors suggest there is a risk involved in using multimodal models (language models conditioned on visual input) and suggest a way to measure this is using a difference of mutual information between a model and the distribution of the training data (is this right?) – though it remains unclear to me how that distribution is gotten.

**Claims And Evidence:**

The abstract talks about the risk of MLLMs and safe and reliable applications, and the proposed framework about quantifying risk under distribution shifts. It’s not clear what this risk is. Would it make sense to just call it the performance?

You talk about multimodal language models in general in the introduction, but it seems this is specifically about (bi-modal) vision and language models with vision only in the input. It could also be made clearer that the paper is only about textual output.

I also don’t see the reason for focusing on vision in the input and instruction tuning. It seems to me the part of the ideas I followed apply to sequence models in general.  As such the question is to what degree a conditional distribution (given some prefix) differs from the non-conditional distribution.

**Essential References Not Discussed:**

Unclear

**Experimental Designs Or Analyses:**

No, since I struggled to understand what is being done

**Methods And Evaluation Criteria:**

I don’t understand how the various probability distributions are defined. I don't understand the definitions of the visual and textual shifts, e.g. the relation between $D(P_{X_v}|| Q_{X_v})$ and $D(P_{X_t}|| Q_{X_t})$. How are these defined exactly in terms of next token probabilities? What is $P_{X_v}$ and $P_{X_t}$? You only define KL with some arbitrary distribution P.

How are $P_X$ and $P_Y$ defined? Are these trained models? Are $P_X,P_Y,P_{\theta},P_{XY}$ all the same trained model differing only in prefixes? E.g. as used in equation 3. For all of these, I would expect to see exact definitions of the distributions in terms of the exact model configuration you say your method works for.

**Other Comments Or Suggestions:**

Some notes:

In the motivation with visual shifts, I would put the definitions of the variables before the shift examples, it was confusing.

Random variables: Could you define the domains of the random variables in line 93? How is (X_v, X_t) combined into a single sequence?

Line 101: instruction tuning has not been introduced.
Line 99: joint population → joint probability?
Equation 1, should this not be argmin?
Equation 2. Should this be P_{\theta} instead of \theta

**Other Strengths And Weaknesses:**

See other answers

**Questions For Authors:**

I think the biggest problem with the paper is the writing and underspecified mathematics. The text is unclear throughout the abstract, introduction, and the motivation of what is being done. I don't find it clear what problem is being solved, nor what the exact configuration in which your method applies is.

More concretely, I don’t find the distributions and the distribution shifts well-defined making it very hard to follow any derivations or reasoning. Even if these (in your mind) are somewhat standard conditional factorizations, I want to see it spelled out, for every such $P_X, P_Y P_{XY}, P_{\theta}$, otherwise it’s very hard to be sure that what is written is correct. It was also unclear what is trained and what is not trained.

The authors define mutual information as a function of a single distribution with an implicit factorization instead of over two random variables. This did not make it easier to follow the writing. For instance, in equation 7 of the EMI (a main contribution), what is the definition of $I(P_X\otimes P_{\theta})$, the definition of $I$ is hardcoded for a distribution $P_{XY}$ based on the factorization given. If there is some marginalization used to define it this needs to be defined, or maybe stick to the mutual information between two random variables.

**Relation To Broader Scientific Literature:**

Unclear.

**Theoretical Claims:**

I struggled to follow the definitions of the authors and so did not go through the proofs.

---

> ### Author Rebuttal · Authors · 2025-03-31
>
> We appreciate sTSP's effort to read our paper and provide comments. Here is a notation table and our responses.
>
> |Var.|Def.|
> |-|-|
> |$X_t=(X_{t,1},...,X_{t,L_t})$ where $X_{t,i}\in V$|a random variable (r.v.) of a text input sequence with length $L_t$ of tokens in vocabulary $V$|
> |$X_v=(X_{v,1},...,X_{v,L_v})$ where $X_{v,i}\in \mathbb{R}^{D_v}$| a r.v. of a $D_v$-dimensional image embedding sequence with length $L_v$ of tokens produced by a frozen vision encoder|
> |$\mathbf{X}=(X_v,X_t)$| a joint r.v. of an input query constructed with a tuple of $X_v$ and $X_t$|
> |$Y=(Y_1,...,Y_L)$ where $Y_i\in V$|a r.v. of a text response with length $L$ of tokens|
> |$P_{\mathbf{X}}=P(X_{v,1},...,X_{v,L_v},X_{t,1},...,X_{t,L_t})$|a probability distribution (p.d.) of an input|
> |$P_{X_v}=P(X_{v,1},...,X_{v,L_v})$|a p.d. of a visual input|
> |$P_{X_t}=P(X_{t,1},...,X_{t,L_t})$|a p.d. of a text input|
> |$P_{Y}=P(Y_1,...,Y_L)$| a p.d. of a text response|
> |$P_{Y \|\mathbf{X}}=P(Y_1,...,Y_L\|X_{v,1},...,X_{v,L_v},X_{t,1},...,X_{t,L_t})$|a conditional p.d. of a text response given input|
> |$P_{\mathbf{X}Y}=P(X_{v,1},...,X_{v,L_v},X_{t,1},...,X_{t,L_t},Y_1,...,Y_L)$|a joint p.d. of input and response|
> |$P_{\theta}(Y\|\mathbf{X})=P_{\theta}(Y_1,...,Y_L\|X_{v,1},...,X_{v,L_v},X_{t,1},...,X_{t,L_t})$|model's prediction p.d. for a response given input|
>
> > What problem is being solved? What is the risk?
>
> * As we noted in the introduction, MLLMs suffer from performance degradation when they encounter distribution shifts. We denote risk as this performance degradation.
> * `Problem focus`: As noted in the abstract, introduction, and motivation sections, **our goal is to quantify the performance degradation of MLLMs under distribution shifts by presenting an information-theoretic framework**.
>
> > It seems this is about (bi-modal) vision and language models with vision only in the input.
>
> * As we noted in `L107-109`, MLLMs take multimodal input (both visual and text) to produce text output, not "vision only in the input".
> * The term MLLM is commonly used in the literature to denote LLMs that receive a visual input as well as text [1,2], so we took this term by following the convention.
>
> > Definition of $P_X,P_Y,P_{XY},P_{\theta}$.
>
> * In `L92-103`, we put the definition of random variables and distributions. $P_X,P_Y,P_{XY}$ denote the probability distributions of the input $\mathbf{X}$, target response $Y$, and their joint $\mathbf{X}Y$, respectively.
> * The $P_\theta$ is a model being trained.
>
> > Definitions of $D(P_{X_v}||Q_{X_v})$ and $D(P_{X_t}||Q_{X_t})$. How are these defined in terms of next token probabilities? What is $P_{X_v}$ and $P_{X_t}$?
>
> As noted in `L96-97`, $X_v$ and $X_t$ are the sequences of visual and text input tokens, so the $P_{X_v}$ and $P_{X_t}$ are the corresponding distributions. $D(P_{X_v}||Q_{X_v})$ and $D(P_{X_t}||Q_{X_t})$ are defined at the input, which should not be confused with the next token probabilities (at the output level).
>
> > Put the definitions of variables before shift examples. Domains of the variables in L93? How is (X_v,X_t) combined into a sequence?
>
> * We put the definition of all variables in `L92-103` before the motivation section.
> * We concatenate the visual input tokens $X_v$ and textual input tokens $X_t$ into a single sequence, where the visual tokens are obtained by encoding an image using a vision encoder (CLIP-ViT), and then projecting them into the language embedding space. This follows the standard practice in MLLMs, where visual tokens are prepended to the text tokens to form a unified input sequence.
>
> > Definition of MI and EMI
>
> * Our main interest is to express the model performance gap across different distributions. Thus, instead of defining MI with individual variables, we define MI as their joint distribution, which is mathematically equivalent to random variable-based MI, as can be seen in Eq 3.
> * The definition of EMI was explicitly introduced in Eq 6. Please refer to `L168-169`.
>
> >  L101: instruction tuning has not been introduced. L99: joint probability? Eq 1, should this not be argmin? Eq 2. Should this be P_{\theta} instead of \theta
>
> * Instruction tuning was introduced from `L104`.
> * In statistics, the population (distribution) [3] is used to denote a distribution of the entire collection of objects in contrast to a sampled distribution.
> * In Eq 1, both min and argmin can be valid, where the former aims at the objective-centric whereas the latter stands for the parameter-centric perspective.
> * In Eq 2, we use the first argument to denote the data distribution that the metric is computed on, and use the second argument to denote the model parameter to be evaluated. **We will use $P_{\theta}$ in the revised version.**
>
> > Reference
>
> 1. A Fully Open, Vision-Centric Exploration of Multimodal LLMs, Tong et al. 2024
> 2. Exploring The Design Space for Multimodal LLMs with Mixture of Encoders, Shi et al. 2025
> 3. Sampling of Populations: Methods and Applications, Levy and Lemeshow 2013

---

### Official Review · Reviewer_o1Fb · 2025-03-21

**Overall Recommendation:** 4

**Summary:**

The paper proposes an information-theoretic framework to analyze the performance of multimodal large language models (MLLMs) under distribution shifts. It introduces Effective Mutual Information (EMI) to quantify the relevance between input queries and model responses. The authors also derive an upper bound for the EMI difference between in-distribution (ID) and out-of-distribution (OOD) data, connecting it to visual and textual distributional discrepancies. Extensive experiments on real benchmark datasets validate the theoretical insights.

**Claims And Evidence:**

The claims made in the paper are supported by clear and convincing evidence.

**Essential References Not Discussed:**

N/A

**Experimental Designs Or Analyses:**

All checked

**Methods And Evaluation Criteria:**

Yes

**Other Comments Or Suggestions:**

N/A

**Other Strengths And Weaknesses:**

- Strengths:  The connection between EMI and win rate provides a practical and efficient alternative for model evaluation.
- Weaknesses: Evaluations are only made on LLaVA v1.5 and LLaVA NeXT. It could benefit from involving the SOTA and representative MLLMs like Qwen2.5-VL and InternVL2.5.

**Questions For Authors:**

- How do the assumptions made in the theoretical framework impact the applicability of the results to more complex real-world scenarios? Can these assumptions be relaxed in future work?
- The paper mentions the potential use of the upper bound of EMID as a regularizer during post-training or test-time adaptation. Can you provide more details on how this could be implemented and its potential impact on model robustness?

**Relation To Broader Scientific Literature:**

The proposed information-theoretic framework for analyzing MLLMs under distribution shifts provided insights for better understanding and alleviating that issue.

**Theoretical Claims:**

The correctness of the theoretical claims, including Lemma 4.3, Theorem 4.4, Theorem 4.5, and Theorem 4.6, is supported by detailed proofs in the supplementary material. The derivations appear to be sound.

---

> ### Author Rebuttal · Authors · 2025-03-31
>
> > _A1. Applicability on SOTA MLLMs._
>
> Thanks for the suggestion! Following your comment, **we additionally conduct the full evaluation with`Qwen2.5-VL-7B-Instruct` and `InternVL2.5-7B`**.
>
> Specifically, we first evaluate the official release of `Qwen2.5-VL-7B-Instruct` and `InternVL2.5-7B` models on 35 synthetic shifts, then compute EMI estimates over the model responses. We perform a correlation analysis between estimates of EMI difference (EMID) and its upper bound. **Consistent with our existing finding, we observe a strong correlation between EMID and its theoretical upper bound in both models**.
>
> |Model|Pearson $r$|$p$-val|
> |-|-|-|
> |InternVL2.5-7B|0.67|0.00|
> |Qwen2.5-VL-7B|0.81|0.00|
>
> > _A2. How do assumptions impact the applicability of proposals to more complex cases?_
> This is an insightful question.
> First, **to claim the closeness between EMI and win rate (Thm 4.4), we assumed $\epsilon$-representation capacity of the MLLM**.
>   * $\epsilon$-representation capacity essentially reflects the model’s ability to approximate the target task’s conditional distribution, meaning that the model can approximate this distribution with a KLD no greater than $\epsilon$.
>   * The assumption mainly argues the strong expressive and approximation capabilities of MLLMs. **Given the strong expressive and approximation capabilities of recent large-scale MLLMs, this assumption is generally reasonable in practice.**
>   * Moreover, numerous efforts improve the diversity of an instruction tuning dataset and robustness of the visual encoder of MLLM [1,2], which makes the learned distribution robustly approximate conditional distributions encountered during evaluation.
>   * As we continually pursue enriching dataset construction and enhancing visual recognition of the encoder, $\epsilon$-representation capacity assumption becomes reasonable in more complex cases.
>
> Second, **to claim the relationship between EMID and its upper bound in a simple case (Thm 4.5), we assumed consistent conditional distributions** over $X_v|X_t$, $X_t|X_v$, and $Y|X$.
>   * This assumption zeros out the discrepancy between conditional distributions. If the conditional distributions are quite different between ID and OOD in some real-world scenarios, this makes our upper bound underestimate the performance gap, e.g., EMID.
>   * However, we highlight that the strong correlations between EMID and this upper bound have been observed through 61 distribution shifts, implying the validity of our upper bound to quantify EMID.
>   * **Meanwhile, we also provide a bound for general cases to address non-consistent conditional distributions in Thm 4.6**. This general-case bound can also be empirically estimated using a procedure similar to that of Thm 4.5.
>   * Therefore, as mentioned in our manuscript (L302-303), we recommend choosing a proper bound based on the knowledge of the data-generating process for datasets.
>
> > _A3. Details for practical implications of EMID upper bound_.
>
> While we confined the scope of this project to _presenting the first theoretical framework to quantify MLLM's performance gap_, we further provide **a potential application of EMID upper bound, instruction tuning with regularization, for this rebuttal**.
>
> Without loss of generality, we can assume the input sequence $X=(X_v,X_t)$ as a sequence of intermediate representation vectors of MLLM, i.e., $Z=(Z_v,Z_t)$, and can also assume that $P_{\theta}(.|.)$ maps this representation to responses, i.e., $P_{\theta}:Z \rightarrow Y$. This induces a modified bound with representation variable $Z$ rather than raw data input $X$.
>
> We instantiate this modified EMID bound in two distinct setups below, where we set the 24th layer's hidden states as $Z$, and adopt RJSD [3] as a differentiable estimator for JSD and an average of empirical model output entropy. We provide evaluation results with LLaVA-v1.5-7B on in-distribution (ID) and visual (V), text (T), and joint (J) synthetic shifts.
>
> `Regularization term for instruction tuning`: $\mathbb{E}[H(P_{\theta}(\cdot|z))] \cdot ((D_{JS}^{0.5}(P_{Z_v}||N(0,I))+(D_{JS}^{0.5}(P_{Z_t}||N(0,I)))$
>   * One can not access $Q_X$ during the training phase, so we alternatively enforce the distribution of the intermediate representation to be close to the standard Gaussian.
>   * We sampled 10% of the instruction dataset from LLaVA-v1.5, and trained the entire LLM and modality connector parameters of LLaVA-v1.5 with and without the regularization.
>   * As shown in the table, we confirm that the EMID can be leveraged as a regularizer during instruction tuning to pursue better robustness to distribution shifts.
> |Method|ID|V Shift|T Shift|J Shift|
> |-|-|-|-|-|
> |Baseline|72.7|65.8|68.0|59.6|
> |Baseline + Ours|72.7|**66.3**|**68.3**|**60.8**|
>
> > Reference
> 1. Qwen2.5-VL Technical Report, Alibaba Group 2025
> 2. Eagle: Exploring The Design Space for Multimodal LLMs with Mixture of Encoders, Shi et al. 2025
> 3. The Representation Jensen-Shannon Divergence, Hoyos-Osorio et al. 2024

---

> > ### Comment · Reviewer_o1Fb · 2025-04-06
> >
> > I appreciate the response from the authors, which addresses my concerns. Thus, I would like to increase my score to Accept.

---

> > > ### Author Response · Authors · 2025-04-06
> > >
> > > We thank the reviewer for taking the time to read our rebuttal and increasing the score. We appreciate your insightful comments and support!

---

### Decision · Program_Chairs · 2025-05-01

**Decision:**

Accept (poster)

**Comment:**

This paper proposes Effective Mutual Information (EMI), a new metric to measure query-response relevance in MLLMs. Unlike standard Mutual Information, EMI eliminates domain-specific elements, enhancing generalizability in out-of-distribution (OOD) settings. With multiple LLaVA variants and benchmarks, experiments confirm EMI’s effectiveness, aligning with theoretical predictions across various distribution shift scenarios.